# Added value of site load measurements in probabilistic lifetime extension: a Lillgrund case study

Shadan Mozafari[1,3,*], Jennifer Marie Rinker[1], Paul Veers[2], and Katherine Dykes[1]

[1]Department of Wind Energy, Technical University of Denmark, Roskilde, Denmark
[2]National Renewable Energy Laboratory (NREL), Golden, CO, USA.
[3]DNV A/S, Tuborg Parkvej 8, Hellerup, Denmark.

**Correspondence:** Shadan Mozafari (Shad.mzf@gmail.com)

**Abstract.** Site-specific fatigue estimation is an essential part of wind turbine lifetime extension, with various methods depending on data availability.

The present study compares probabilistic lifetime extension assessment results for rotor blades with and without load measurements. It also addresses two key questions in such assessments: the applicability of the Frandsen model for estimating waked turbulence under complex and mixed wake conditions, and the extrapolation of mid-term data over longer time periods.

The case study wind turbine is SWT-2.3-93 located at the edge of the Lillgrund wind farm, situated in the Øresund Strait between Denmark and Sweden. The turbine is extensively instrumented, with five years of data available from its Supervisory Control (SCADA) system.

Although the Frandsen turbulence estimates deviate in a different manner from measurements at below- and above-rated mean wind speeds, the model remains a conservative approach for fatigue load prediction and reliability.

In the current case study, the site-specific assessment using strain gauge measurements yields a 33% higher annual fatigue reliability index after 35 years compared to a scenario based on the Frandsen estimation combined with ambient environmental data, and a generic aeroelastic model. The results also demonstrate that the sensitivity of fatigue reliability to load uncertainty is negligible when load measurements are used directly, but relatively high when relying on the Frandsen model in combination with a generic aeroelastic model. Overall, the high variability of the lifetime extension in different scenarios of data availability and accuracy shows the importance and added value of high quality measurements combined with wind farm level SCADA and model updated in real-time (digital twins).

**keywords**: Frandsen model, Lifetime extension, Digital twin, Fatigue reliability, Wind farm wakes, Statistical extrapolation, Wind turbines

## 1 Introduction

Exposure of wind turbines to the wakes within a wind farm increases fatigue loads they experience (Kim et al. , 2015; Lee et al. , 2013; Frandsen , 2007). However, the design assumptions in IEC 61400-1 standard are often conservative enough to seek an extension of the operation time even after experiencing the high fatigue loads caused by wakes in the wind farms. Extending

the operation time above the design service life (typically 20–25 years), in cases where maintaining safety is possible, is environmentally beneficial and can reduce the levelized cost of energy (Dimitrov and Natarajan , 2020; Natarajan et al. , 2020).

When it comes to lifetime extension of wind turbines in a wind farm, one must reassess the service lifetime by replacing the design assumptions with the conditions experienced at the site as the analytical part (DNV-ST-0262 , 2016). In such reassessments, fatigue is typically the primary focus, as it represents one of the most critical time-dependent degradation mechanisms. Information about in-situ lifetime can be obtained in various ways, depending on data availability. Often, the turbine model is not available and generic models are used. Some of the common scenarios under this condition, include the following:

1. Often, high-quality environmental measurements at the turbine's specific location are not available, whereas freestream turbulence measurements are. In such cases, waked turbulence at each turbine location can be estimated using simplified models such as the Frandsen model (Frandsen , 2007; Frandsen and Madsen , 2003), suggested by IEC 61400-1 (2019) for site suitability checks. The resulting estimates are then used to perform aeroelastic simulations with a generic turbine model, from which site-specific lifetime can be assessed.

2. In some cases, structural response measurements (e.g., load or displacement) are available for a limited duration and at specific hotspot locations. Direct use of these measurements for lifetime extension assessment is possible, but it involves challenges such as spatial extrapolation (from one location in the structure to another) and temporal extrapolation (from one point in time to another).

3. When structural response measurements are complemented by Supervisory Control and Data Acquisition (SCADA) data and high-quality environmental measurements, a digital twin can be developed to estimate loads accurately in all components or locations within the turbine.

Often, structural response measurements collected at the site are owned by the turbine manufacturer and are not accessible to the wind farm owner or developer. Moreover, when available, such measurements are typically limited in both duration and spatial coverage. This research compares lifetime extension assessments for two scenarios, using a case study in which both scenarios can be explored. Because of the inaccuracy of the wind measurements from the anemometer -due to the location of the met-mast and unavailability of the SCADA of the neighboring wind turbines- the measurements are not directly used in waked directions. Free-stream measurements are used for model validation (see appendix). In addition, the study addresses two common challenges encountered in these scenarios, as outlined below:

1. Evaluating the performance of the Frandsen model, a simplified method for estimating wake-induced turbulence, in a compact wind farm layout (scenario 1).

2. Developing a method for the statistical extrapolation of mid-term strain gauge data to estimate long-term fatigue loads (scenario 2).

The Frandsen model is based on simplified assumptions and carries inherent uncertainties. The uncertainty associated with fatigue load estimation using this model is discussed in a few examples by Frandsen (2007). Several limitations of the Frandsen model have also been identified. For instance, Bayo and Parro (2015) and Argyle et al. (2018) highlight issues such as the lack of a defined method for accounting for wake interactions among multiple turbines and the potential for under-conservative predictions when the wind farm significantly affects the mean wind speed. Therefore, in wind farms with compact and/or irregular layouts, verifying the Frandsen model's performance for conservative site-suitability assessments is crucial. Although prior studies have examined the model's general performance, its applicability in high wake-interaction conditions—particularly those resulting from tight turbine spacing (e.g., less than five rotor diameters) and irregular configurations—remains unexplored. The current work compares turbulence estimates from the Frandsen model with actual measurements at the Lillgrund wind farm, as an example of compact layout, to assess the accuracy of the model under various waked flow conditions.

A challenge in scenario 2 for assessing site-specific lifetime is the limitation of available data and the need to extrapolate short-term load measurements to cover longer periods for fatigue damage estimation. Determining turbine-specific fatigue accumulation throughout the operational lifetime under such constraints remains a critical open question. This issue is particularly significant for offshore wind farms, where the high variability and scatter in environmental conditions adds to the complexity of accurate fatigue lifetime assessment. Some studies, including Amiri et al. (2019); Ziegler and Muskulus (2016), assess lifetime extension by assuming a linear increase in fatigue damage over time. However, this approach can introduce significant errors when the available data are insufficient to accurately estimate the Damage Equivalent Load (DEL). Although DEL is an averaged metric and therefore generally more robust to individual outliers (Mozafari et al. , 2023b), it can still exhibit considerable variability depending on the length of the dataset—particularly for components with high fatigue exponents. As an example, the results of Mozafari et al. (2023a) show that the conventional approach of assuming a constant DEL—implying a linear increase in damage accumulation over time—can introduce high bias in long-term fatigue damage assessments, particularly for blades. This is due to the high fatigue exponent associated with composite materials used in blade structures.

Some studies, such as Dimitrov and Natarajan (2019) and Natarajan et al. (2020), have employed machine learning techniques and Monte Carlo simulations to generate long-term fatigue load estimates from mid-term response measurements. Other studies, including Ling et al. (2011), Hübler et al. (2018), and Natarajan (2022), have applied stochastic methods to predict long-term fatigue behavior based on shorter-term load data. Despite different statistical approaches and the guidelines given in IEC 61400-1 (2019), the best procedure to statistically extrapolate fatigue loads remains unclear. The current work presents a procedure capturing multimodality of fatigue load data—as demonstrated to be effective in Mozafari et al. (2023a)—and extrapolates it over the full assessment duration.

The current study assesses the feasibility of extending the operational lifetime of the wind turbine blades by 10 years at the edge of the Lillgrund wind farm, while ensuring that acceptable safety margins are maintained. In the current study, the turbulence data in a location close to the wind turbine is available and is utilized for validation purposes and to illustrate the extent of error associated with the generic model used in the study.

First, demonstrating scenario 1, using the available SCADA data and the generic model of the turbine, we evaluate the performance of the Frandsen model in conservatively estimating fatigue loads at the case study wind farm. To enable a more

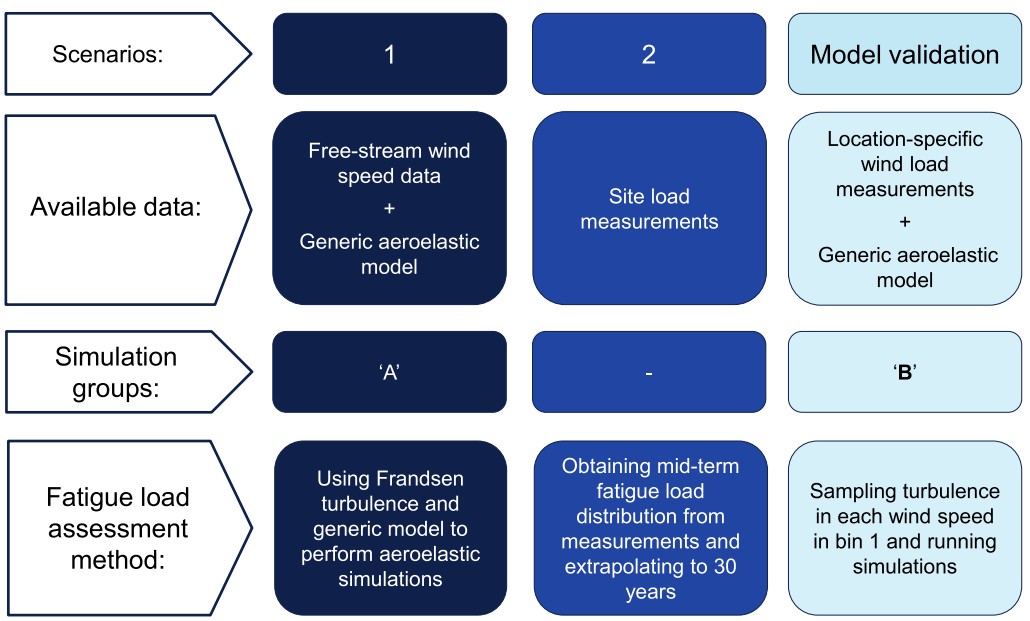

**Figure 1.** Overview of site-specific assessment scenarios (studied in the present work) with corresponding data availability, simulation approaches, and load estimation methods

detailed assessment of the model's turbulence predictions, we classify and bin different wake scenarios affecting the case study wind turbine. Second, demonstrating scenario 2, we employ a Gamma mixture model to represent the bi-modally distributed mid-term Damage Equivalent Load (DEL) data, derived from 10-minute strain measurements at the blade root, and extrapolates them to a 30-year operational lifetime. Finally, we show the annual fatigue reliability levels of the blade in different scenarios versus design (simulated via the generic model) to highlight potential differences in the lifetime extension feasibility. In addition, the relative influence of three key factors on fatigue reliability estimation across scenarios 1 and 2 is studied: the applied loads, the material's fatigue strength, and the chosen damage accumulation rule. Different scenarios and the corresponding simulations used in the current study are shown in figure 1.

Awareness of the effects of different sources of uncertainty on lifetime extension assessment is valuable and can help improve the accuracy and robustness of such evaluations. The presented procedure for extrapolating fatigue loads can help stakeholders and wind farm owners obtain a more accurate assessment of fatigue damage in cases where strain gauge measurements are unavailable or only available for a limited period of the turbine's lifetime. To make the life extension results interesting for this specific example, the material properties are calibrated to achieve a target reliability index of 3.7 (IEC 61400-1 , 2019) after 20 years, based on the design class (as would result from applying the recommended safety factors from the standard). Thus, while the comparisons of reliability levels are valid, their absolute magnitudes are not.

In the next sections, first, we present the methods and models we use for modeling and reliability assessment (Sect. 2). Then we present the results and corresponding discussions of the Frandsen performance check and lifetime extension assessments in Sect. 3. Finally, in Sect. 4, we present the conclusions of both studies and suggestions for future work.

## 2 Methodology

First, we introduce the case study wind turbine and the corresponding wind farm in Sect. 2.1 and Sect. 2.2, respectively. Then, in Sect. 2.3, we present the setup and features of the aeroelastic simulations and site load measurements. In addition, we introduce the methods we use for assessing and filtering data for the current study. Finally, in Sect. 2.4, we introduce the mathematical formulas and the procedures we use for post-processing the simulation results and load measurements.

### 2.1 The case study wind turbine

The wind turbine under study in the current research is SWT-2.3-93, manufactured by Siemens Energy. The turbine has a 92.6 m rotor diameter, a hub height of 65 m, and a rated power of 2.3 MW. The cut-in and cut-out mean wind speeds are 3 $ms^{-1}$ and 25 $ms^{-1}$, respectively, reaching the nominal power at approximately 12–13 $ms^{-1}$. The turbine belongs to class 1A based on the IEC 61400-1 standard's classification. A generic model of this turbine is used for the assessments.

### 2.2 The case study wind farm

The strain gauge and environmental measurements belong to one of the turbines at the edge of the Lillgrund offshore wind farm. Lillgrund wind farm is located about 10 km off the coast of Sweden in the Öresund region and consists of 48 Siemens SWT-2.3 93 wind turbines (total capacity of 110 MW). The turbines are arranged as shown in Fig. 2. The circles in Fig. 2 represent different turbines, and the red circle is the case study wind turbine, denoted as C08 (row c and column 8). We bin the wind directions around the case study wind turbine to roughly distinguish between different wake conditions. The binning facilitates assessing the performance of the Frandsen model and IEC NTM assumptions in characterizing turbulence in different wake scenarios. The dashed lines in Fig. 2 show the bins. For the rest of the study, we refer to the wind direction bins as "wind bins" for simplicity.

As Fig. 2 illustrates, wind bin 1 accounts for non-waked conditions. In wind bins 2, 3, and 7, the stream is mainly passing by a single turbine, with bin 3 representing a relatively long distance between the turbine generating the wake and the C08 wind turbine. Wind bin 5 is an intense condition, with the case study turbine being very deep in the row and a very short distance between the closest wind turbine (4.3 times the rotor diameter). Bin 4 is a similar arrangement of a single row with a relatively long distance, and bin 6 is a mixed-waked flow condition. One should note that the binning and the corresponding conditions described are qualitative and are only used for general comparison.

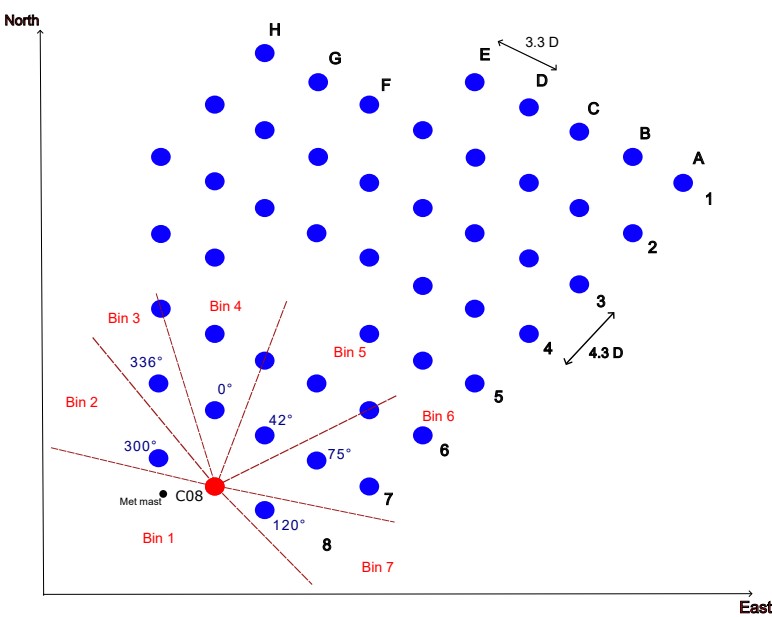

**Figure 2.** Arrangement of the wind turbines in the Lillgrund wind farm and the wind direction bins around the turbine C08 used in the current study

## 2.3 Measurement data and aeroelastic simulations

In the present study, we use SCADA and 10-minute DEL evaluations extracted from a strain gauge installed at the blade root.
The data represent long-duration measurements over a span of 5 years in the Lillgrund wind farm. We also perform aeroelastic simulations in HAWC2 software (Larsen and Hansen , 2007). In the following, first, we introduce the available measurement data (from the strain gauge and SCADA). Then, we present the specifications of the three groups of aeroelastic simulations used.

### 2.3.1 Measurement data

The available data include strain gauge measurements and SCADA records, comprising wind speed, wind direction, power output, and rotor speed. The mean and standard deviation of wind are derived based on an anemometer mounted on a meteorological mast located in close proximity to the case study wind turbine. (black dot in Fig. 2). The meteorological mast is placed on a pole on top of the tower at the height of 65 m (for more information see Bergström, H. (2009)). In addition, the strain gauge is installed at 1.5 m away from the blade root. All data are measured in time intervals of 10 minutes during years 2008 to 2012. The strain gauge measurements are transformed into 10-minute DELs using the Palmgren-Miner approach, considering a Wöhler exponent equal to ten for the composite structure of the blade. The measurement campaign has been running over 5 years but not continuously.The data spans approximately two years in total duration. However, it is distributed across three

months in 2008 and all months of 2009, 2010, 2011, and 2012. As a result, some measurements include events with a return period of five years. Although the non-operational conditions should also be considered for fatigue assessments, in the current study, we only focus on normal operating conditions in the simulations (DLC 1.2 in the IEC standard). We use the provided DEL data to estimate the site-specific fatigue loads for the comparison study after filtration. We use the power curve of the turbine and utilize the data from SCADA, including mean wind speed, mean rotor speed, and the mean power production for the same points of time to filter the corresponding DEL data. Thus, the wrong measurements and the ones that do not represent the operational conditions are omitted (Fig. C1 in Appendix shows the power curve after such filtration). A total of 88031 data points remain after the filtration.

Table 1 shows the normalized number of data available in different wind direction bins around the C08 wind turbine (see the bins in Table 2) after filtration. In addition, Table 1 shows the probability of occurrence of each bin based on the wind rose of the site Vitulli et al. (2019).

**Table 1.** Binning of wind directions and their corresponding probability in the Lillgrund wind farm site based on Vitulli et al. (2019)

| Bin number | Wind direction bounds (degrees) | Probability (%) | Normalized number of available data in 5 years (%) |
|---|---|---|---|
| 1 | 135-285 (non-waked) | 55.5 | 56.56 |
| 2 | 285-315 | 9.6 | 7.41 |
| 3 | 315-345 | 5.0 | 7.15 |
| 4 | 345-360, 0-15 | 5.4 | 2.98 |
| 5 | 15-60 | 6.9 | 6.23 |
| 6 | 60-105 | 9.3 | 12.61 |
| 7 | 105-135 | 8.3 | 8.51 |

As Table 1 illustrates, the available data after filtration do not fully cover the wind direction probabilities. This is partly because in some heavily waked conditions the turbine has been shut down and partly because of the short duration of the data gathering. In the current work, we keep the mentioned observation in mind as a limitation. Thus, we proceed by assessing all DEL data collectively for the purposes of fitting, extrapolation, and bootstrapping. Fig. C3 (in appendix) shows the separate analysis of the DEL data in each wind direction bin. This analysis ranks each bin based on both the highest observed and expected DEL values. Such ranking should be considered when performing fatigue assessments per bin to ensure more accurate evaluations. However, in this study, we estimate damage accumulation based on the overall dataset. This limitation is further discussed in Sect. 4.

In the current case, different distributions best describing the turbulence in each wind speed in the freestream are used. However, we do not present the details of those fits to be concise. We sample from the distributions and use them as input for aeroelastic simulations in the second part of the study (see Sect. 2.3.2).

## 2.3.2 Aeroelastic simulations

As mentioned before, a generic model of SWT-2.3 turbine is used in the present study. In this model, the structural and aerodynamic properties were supplied by Siemens Energy, whereas the controller is a tunned version of the DTU 10MW controller. We conduct all simulations using this model. The benchmark simulations are based on inputs from IEC 61400-1 IEC 61400-1 (2005) requirements for design of wind turbine class A. Group 'A' is based on the IEC 61400-1 recommendation for site suitability check (i.e., the Frandsen model, explained in Sec. 2.4). Group 'B' is based on the site-specific inputs in terms of turbulence and wind shear exponent from the freestream (wind bin 1 in Fig. 2). This group of simulations is used solely for the validation of the generic HAWC2 turbine model. The validation results are provided in the appendix.

Groups A and B have 11 mean wind speed bins (from 4 $ms^{-1}$ up to 24 $ms^{-1}$ with 2 $ms^{-1}$ intervals). Group B only includes the mean wind speeds up to 20 $ms^{-1}$ as the number of data points and probability of occurrence for the higher mean wind speeds are very low. The two first groups of simulations have one representative turbulence level in each mean wind speed bin. This level equals the 90% percentile value in group A and the Frandsen waked turbulence level in group B (more details are provided in the next section).

We consider 100 seeds for each wind condition to account for the variability of the wind Mozafari et al. (2023b) and have enough data for fitting distributions to the resulted DELs. The third group includes 20 samples from the site-specific turbulence standard deviation distribution in each mean wind speed in the freestream directions (wind bin 1). In this group, having nine mean wind speeds in the related wind directions, and 100 turbulence seeds for each turbulence level resulted in a total number of 22000 simulations of 10 minute duration for the site-specific aeroelastic simulations (group B). Considering a lognormal wind profile in all simulations, we consider the shear exponent equal to 0.2 in the first two groups and equal to 0.1 in group B as an estimation for shear exponent in smooth terrain (open water) Yan et al. (2022). This value is also close to the lidar measurements in the site Liew et al. (2023). We use the Mann turbulence model Mann (1998) to generate the simulation turbulence boxes. The boxes contain 8192 evaluation points alongside the wind direction for higher resolution and 32 points in the other two directions. Table 2 shows the specifications of each group of aeroelastic simulations.

It should be noted that Group 1 is based on Edition 1 of the IEC standard, which represents the common design framework used for the previous generation of wind turbines—and is therefore particularly relevant for lifetime extension assessments today. Alternatively, when a full lognormal or Weibull distribution is used—corresponding to Editions 3 and 4, respectively—the results of the study differ. (see Mozafari et al. (2024) for differences).

The following section outlines the mathematical formulations and procedures used in the present study.

## 2.4 Mathematical formulations

In the current section, first, we introduce the wind characteristics, and then we briefly present the relations for fatigue load assessment plus methods for the reliability and importance ranks. Finally, we explain the procedure for forming the database for statistical extrapolation of DEL measurements via bootstrapping.

**Table 2.** Specifications of wind modeling in three groups of HAWC2 simulations corresponding to three study cases

| Parameter | Benchmark (design) | Group A (Frandsen) | Group B (Validation) |
|---|---|---|---|
| Turbulence in each MWS | 90% quantile in NTM | Frandsen's effective turbulence | Ambient turbulence distribution (freestream bin) |
| Reference turbulence intensity | 0.16 | 0.11 | - |
| Wind shear exponent | 0.2 | 0.1 | |
| Turbulence levels in each MWS bin | 1 | | 20s |
| Realizations per wind condition | 100 | | |
| Turbulence model | Mann | | |
| Cut-in MWS ($m/s$) | 3 | | |
| Cut-out MWS ($m/s$) | 25 | | |
| Rated wind speed ($m/s$) | 11.4 | | |
| Size of wind speed bins ($m/s$) | 2 | | |
| Yaw misalignment (degrees) | 0 | | |
| Mann box grids along the wind | 8192 | | |
| Mann box grids in other dimensions | 32 | | |
| Mann turbulence length scale | 29.7 | | |
| Mann turbulence anisotropy factor | 3.7 | | |
| Simulation length (s) | 700 | | |
| Transient time (s) | 100 | | |
| Time steps of the simulations (s) | 0.01 | | |

### 2.4.1 Probabilistic modeling of wind

In the current study, we only observe two random parameters of the wind field: mean wind speed and the wind speeds' standard deviation (i.e., turbulence). Following the IEC standard IEC 61400-1 (2019), we assume the distribution of the mean wind speed at hub height to be Rayleigh for both cases of design-level assessment and site-suitability check (the Frandsen model). In the latter, the mean wind speed in the waked area is assumed to be the same as the freestream. For site-specific assessments under non-waked conditions, the measured distribution of wind speed is used directly, as it more accurately reflects the actual conditions at the site. In the case of design-based assessment, we use 90% quantile of the lognormal distribution as suggested by the normal turbulence model in IEC 61400-1 (2005) as the representative turbulence. Equation (1) presents this level.

$$\sigma_{rep.design} = I_{ref}(0.75 v_{hub} + 5.6) \tag{1}$$

In Eq. (1), $I_{ref}$ is the reference turbulence intensity equal to 0.16 for the standard class 1 wind turbines (the current case study). In addition, $V_{hub}$ is the hub height wind speed.

In the Frandsen model, the freestream standard deviation is assumed to be the 90% quantile of a normal distribution Argyle et al. (2018). In the waked conditions, the turbulence is described as a function of the thrust coefficient and the normalized distance of the closest wind turbine. Equation (2) and Eq. (3) present the freestream standard deviation formulations and enhanced turbulence due to wakes in the Frandsen model.

$$\sigma_{rep.Frandsen} = \mu_\sigma + 1.28 * \sigma_\sigma \tag{2}$$

In Eq. (2), $\sigma$ is the turbulence standard deviation (turbulence) of the freestream wind (ambient flow) considered as a random variable. In addition, $\mu_\sigma$ and $\sigma_\sigma$ refer to the mean and standard deviation of the turbulence, respectively. In scenario 1 of the present study, these values are obtained from measurements corresponding to the freestream wind direction bin.

$$T_{waked}(\theta) = \sqrt{\frac{v_{hub}^2}{\left(1.5 + 0.8\left(\frac{di(\theta)}{\sqrt{C_T}}\right)\right)^2} + \sigma_{rep.Frandsen}^2} \tag{3}$$

In Eq. (3), $\theta$ is the (wind) direction in which the waked turbulence is estimated, $d_i(\theta)$ is the distance of the closed turbine in that direction and $C_T$ is the characteristic value of the wind turbine thrust coefficient for the corresponding hub height wind velocity IEC 61400-1 (2019).We use the thrust coefficient data provided in Montavon et al. (2009) for the current study.

The Frandsen model's turbulence is the same in all wind directions (like in IEC design assessments). This independence from wind direction is obtained by effective turbulence. Equation (4) shows the effective turbulence used for site-suitability check IEC 61400-1 (2019).

$$T_{eff}(V_{hub}) = \left(\sum_{\theta=0}^{2\pi} P_\theta(V_{hub})(T_{waked}(\theta))^m\right)^{\frac{1}{m}} \tag{4}$$

In Eq. (4), $P_\theta(V_{hub}$ is the probability of occurrence of the hub height mean wind speed in each direction ($\theta$), and $m$ is the fatigue (Wöhler) exponent Basquin (1910)[1].

### 2.4.2 DEL estimation

In the present research, we use the Basquin relation Basquin (1910) for modeling the fatigue resistance of the composite material and Palmgren–Miner (Miner's) rule Palmgren (1924); Miner (1945) for modeling the damage accumulation. These models describe the lifetime and damage as functions of stress, while the outputs of the aeroelastic simulations that we use are flapwise bending moments in the blade root. Since the location of interest (the strain gauge installation location) is close to the root and nearly circular, we use Eq. (5) to obtain the stresses based on the moment time series.

$$S_i = \frac{M_{x_i} \times c}{I_y} \tag{5}$$

---

[1]For further details and derivation of the equations 3 and 4, see Frandsen (2007) and IEC 61400-1 (2019).

In Eq. (5), $M_{x_i}$ denotes the bending moment corresponding to the stress level $S_i$. We evaluate the moments ($M_x$) accounting for flapwise moments in the blade root. The section parameters $c$ and $I_y$ represent the local radius (half the chord length) and the second moment of inertia about the axis perpendicular to the moment direction, respectively. Due to confidentiality constraints, the specific values of these parameters are not disclosed.

Using rainflow counting Endo et al. (1967) of the moments in each 10-minute simulation and the models mentioned (Basquin and Palmgren–Miner), we estimate the 10-minute fatigue damage via Eq. (6).

$$D = \left(\frac{c}{I_y}\right)^m \sum_{i=1}^{N_s} \frac{n_i \times (M_{x_i})^m}{k} \tag{6}$$

In Eq. (6), $N_s$ is the total number of cycles at the time frame considered for calculation of the damage. In addition, $m$ is the fatigue exponent and $k$ is the Basquin coefficient (see Basquin (1910)). Reformulation of Eq. (6) using the concept of DEL (see Mozafari et al. (2023b) for more information) results in Eq. (7). We use this expression in the current study to simplify the comparisons.

$$D = \frac{N_{eq}(DEL_{lifetime}^m)}{k} \left(\frac{c}{I}\right)^m \tag{7}$$

In Eq. (7), $N_{eq}$ is the reference number of cycles. We set $N_{eq}$ equal to 600 cycles corresponding to the average of 1 Hz cyclic loading. In addition, $DEL_{lifetime}$ (the expected value of the fatigue damage equivalent load through lifetime) can be derived from 10-minute DEL estimations via Eq. (8).

$$DEL_{lifetime}^m = \sum_{\Theta=\Theta_L}^{\Theta_U} \sum_{V_\Theta=V_L}^{V_U} \sum_{T_{(\Theta,V)}=T_L}^{T_U} E[(DEL_{10min,\Theta})^m] \mathrm{P}(T,V|\Theta)\mathrm{P}(\Theta) \tag{8}$$

In Eq. (8), the parameters $\Theta_L$ and $\Theta_U$, represent the lower and upper bound of wind direction bins, respectively. Similarly, $V_L$ and $V_U$ denote the lower and upper bounds of the mean wind speed, while $t_L$ and $t_U$ correspond to the lower and upper bounds of turbulence intensity within each wind direction bin. Furthermore, $\mathrm{P}(T,V \mid \Theta)$ denotes the joint probability of turbulence intensity and mean wind speed within a given wind direction bin $\Theta$. Since we consider the marginal probability of turbulence conditioned on mean wind speed, $\mathrm{P}(T \mid V)$, and the probability of mean wind speed conditioned on wind direction, $\mathrm{P}(V \mid \Theta)$, the joint probability can be expressed as the product of these conditional probabilities.

In assessments where a constant turbulence level is assumed for each mean wind speed (i.e., Frandsen effective turbulence and IEC representative turbulence characterizations), the probability of the single turbulence value is set to 1. In contrast, for Group 3 simulations, the probability distribution of turbulence within each wind speed bin is explicitly accounted for, while the probability of the wind direction bin is set to 1, since only wind direction bin 1 is considered for model validation. In measurement-based assessments, the lifetime DEL is estimated using the unweighted average of $DEL_{10\text{min}}$, under the assumption that the dataset is sufficiently large for the underlying wind condition probabilities to be implicitly captured. In other words, the distribution of 10-minute DEL values inherently reflects the probability distribution of the wind conditions.

Equation 9 defines the relationship between the available $n$ values of $DEL_{10\text{min}}$ and the corresponding lifetime DEL, $DEL_{\text{lifetime}}$.

$$DEL_{lifetime}^{m} = \sum_{i=1}^{n} \frac{(DEL_{10min_i})^m}{n} \tag{9}$$

In Eq. (9), in the case of DEL in one year, $n$ would be the number of 10 minutes within the timeline of 1 year. If enough $DEL_{10min}$ data are not available, one has two options: statistical extrapolation (as in the current study) or assuming that the same observations repeat during the longer times (for reference to the importance of statistical extrapolation in estimation of $DEL_{lifetime}$, see Mozafari et al. (2023a) and Mozafari et al. (2023b)).

**2.4.3    Forming the DEL database based on measurements and statistical extrapolation**

For the site-specific (measurement-based) assessment of reliability, we need to obtain the distribution of the $\log(DEL_{lifetime})$ in each year to put in Eq. (14) to be able to estimate the annual reliability level up to year 30 using Eq. (16). We prepare the data for such assessments for up to 30 years. Below steps show the procedure:

1. Fitting a distribution to the 10-minute DEL measurements.

2. Extrapolating the fitted distribution from Step 1 to estimate higher quantiles representative of a 30-year return period (Eq. 10 to 12).

3. Taking 500 (more than sufficient according to Mozafari et al. (2023b)) random samples of size $365 \times 24 \times 6 \times N$ from the database with replacement, where N accounts for number of years and repeating from year 1 to 30.

4. Calculating the mean of $(DEL_{10min})^m$ in each of the 500 samples and estimating the corresponding $DEL_{lifetime}$ based on Eq. (9).

5. Calculating the logarithm of all generated data and fitting probability distribution to the 500 realizations of $\log(DEL_{lifetime})$ in each year.

For forming the database (step 2 above), first, we find the probability of exceedance corresponding to 30-year return period using Eq. (10) and Eq. (11). The extrapolation is used to complete the tail of the $DEL_{10min}$ distribution to account for highest values that might change the weighted mean value ($DEL_{lifetime}$) if included. As we are aiming at adding these low probability high magnitude occurrences, extreme value theory can be a suitable model to use. These values can have high effect due to the high fatigue exponent of the composite Mozafari et al. (2023b)

$$CDF(L_R) = \exp\left(\frac{-1}{T_{L_R}}\right) \tag{10}$$

$$Pr_{exc.}(L_R) = (1 - CDF(L_R)) \tag{11}$$

In Eq. (10) and Eq. (11), CDF accounts for cummulative distribution function, $L_R$ accounts for the return load level. Aditionally, $T_{L_R}$ in Eq. (10) accounts for the ultimate time of interest for which the corresponding load is estimated. Equation (10) is extracted from the formula of probability of exceeding a threshold level (here the load with frequency of occurrence of every 30 years) assuming a Poisson process for describing the peaks over threshold problem (for further information see de Oliveira JT (2013)). In the current case, the frequency of exceedance is $1/T_{L_R}$. It has to be noted that Eq. (10) is correct when $T_{L_R}$ is relatively large (here, equal to the number of 10 minute occurrences in 30 years). We set the time in terms of the number of 10 minutes because we consider the DEL to be the load, and in this case, each DEL is an occurrence of a 10-minute duration. The $Pr_{exc.}(L_R)$ in Eq. (11) is the probability of exceeding such load, meaning the probability that a load higher than that level occurs. In the case of return loads, this probability is normally very low. We use the CDF corresponding to the return period, obtained from Eq. (10), to find the return load in our case. This load can be derived by finding the inverse CDF of the distribution of our 10-minute data (step 1 above). After finding the higher load, we can find the number of occurrences of each DEL level in our database based on Eq. (12). In other words, first, the loads with a certain reference return period are defined, and the frequency of lower loads is derived accordingly.

$$i = \frac{1}{\Pr(L_R)} \tag{12}$$

In Eq. (12), $i$ is the number of occurrences of each 10-minute DEL level ($DEL_{10min_i}$*), and $Pr(L_R)$ is the probability of occurrence of the return load based on the distribution of $DEL_{10min}$s (distribution in step 1 above). Equation (12) is based on the assumption that the probability density functions of $DEL_{10min}$ remain the same when more observations are added to the tail through time. The more data involved in fitting the distribution of $DEL_{10min}$, the more accurate this assumption is.

### 2.4.4 Fatigue reliability estimation

Fatigue reliability assessment is performed to obtain an estimation of the probability of the survival of a structure. Equation (13) presents a mathematical representation of this concept.

$$R(t) = 1 - P_f(t) \tag{13}$$

In Eq. (13), $P_f(t)$ is the probability of failure at time $t$. Commonly, this problem is referred to with a function named limit state function ($g(x,t)$), and the safe region is where this function is positive. Thus, the probability of failure would be the probability of lying in a region in the space of random variables where the limit state function is negative or equal to zero. We fully follow the methods and procedures used in our previous work Mozafari et al. (2024) for reliability estimation. Here, we present a brief introduction and the general approach. We recommend the reader to check Mozafari et al. (2024) for further

details.

Following Miner's rule, failure can happen when damage is higher than a threshold level (commonly 1). This rule contains uncertainty due to simplified assumptions like linear damage accumulation without sequence effect. To account for the uncertainty in Miner's rule, we assume the threshold ($\Delta$) as a random variable with a mean value equal to 1. Thus, the reliability being the probability of survival, would be the probability of damage being less than $\Delta$. Equation (14) describes the limit state function considering $DEL$, $K$, and $\Delta$ as random inputs for the case of flapwise bending moments in the blade root with a circular cross section and structural properties of $I_y$ and $c$ (moment of inertia and radius of the cross section, respectively). The time is omitted from Eq. (14) for simplicity with the assumption that all variables are referring to a certain time ($t = lifetime$).

$$g(X, lifetime) = \log(\Delta) - \log(N_{eq}) - m \times \log\left(\frac{c}{I_y}\right) + \log(K) - m \times \log(DEL_{lifetime}) \tag{14}$$

The parameters $log(N_{eq})$ and $m \times log(\frac{c}{I_y})$ in Eq. (14) are constants. Thus, the above equation consists of three random parameters:

   – The linear damage accumulation model ($\log(\Delta)$)

   – Material resistance ($\log(K)$)

   – Load ($\log(DEL_{lifetime})$)

We perform the probabilistic reliability assessment using first-order reliability method (FORM) in the current work to find the probability that the function $g$ in Eq. (14) can be positive (fatigue reliability). The same approach also provides the importance rank of the random variables (sensitivity of the reliability to each) and is used here (see Mozafari et al. (2024) for details and formulations). We use the reliability index (shown in Eq. (15)) as a commonly used measure of structural reliability.

$$\beta = -\Phi^{-1}(P_f) \tag{15}$$

The operator $\Phi^{-1}$ shown in Eq. (15) corresponds to the inverse CDF of the standard normal distribution.

We consider the stress ratio $R = 10$ for fatigue properties (SN curve) of the composite (Mikkelsen , 2020). Although the variability of data is included as the coefficient of variation (CoV) of such curve, a calibration is added at the end to set the mean value of material strength to a certain level at which target level of reliability is obtained at year 20.

To apply FORM analysis, first, we fit distributions to the estimations of $\log(DEL_{lifetime})$ calculated based on 10-minute simulations using Eq. (9) for the measurement-based assessment and Eq. (8) for the simulation-based scenarios. For modeling the uncertainty in the model and material properties in the current study, we gather information regarding the distributions and statistical parameters from the literature. Table 3 shows this information plus the references for the coefficients of variation. The mean of the material resistance is found through calibration. The calibration process entails finding a resistance mean

value for which the target reliability is achieved at the end of the design lifetime.

**Table 3.** Characteristics of the random variables

| Variable | Distribution | Mean | CoV | Reference |
|---|---|---|---|---|
| $log(\Delta)$ | Normal | -0.1116 | 0.31 | Toft and Sørensen  (2011); Toft et al.  (2016) |
| $log(K)$ | Normal | Calibrated ($\beta = 3.7$ at year 20) | 0.528 | Fraisse and Brøndsted  (2017) |
| $log(DEL_{lifetime})$ | GEV | Confidential | 0.001 | Sect. 3.2 |

Equation (16) and Eq. (17) present the formulations for calculating the probability of failure and reliability index at time $t$
($P_f(X,t)$ and $\beta(X,t)$, respectively) conditional on survival in the previous point of time $(t - \Delta t)$ considering a time interval of $\Delta t$. The parameter $X$ is a vector of all random variables (the above-mentioned parameters for the current case study). For further information regarding derivation of Eq. 16 and Eq. 17, see Faber, M. H.  (2012).

$$\Delta P_f(X,t) = \frac{P_f(X,t) - P_f(X,t-\Delta t)}{(1 - P_f(X,t-\Delta t))} \qquad (16)$$

$$\Delta\beta(X,t) = -\Phi^{-1}(\Delta P_f(X,t)) \qquad (17)$$

In the current work, we consider $\Delta t$ to be equal to one year and thus the parameters $\Delta P_f(X,t)$ and $\Delta\beta(X,t)$ correspond to the annual probability of failure and annual reliability index and will be referred to as such in the in the continuing discussion. It must be noted that the Frandsen and IEC turbulence models together with partial safety factors are intended for semideterministic design and not for probabilistic design and reliability analysis. However, since the partial safety factors are calibrated based on achieving a certain reliability level (to which we are also setting the values for) at the end of the design
lifetime, the results are comparable. Such comparisons are presented in the next section.

## 3   Results and discussions

We validate the model of the turbine before performing the study (see Appendix for validation results). The current section presents the results in three parts: Turbulence comparison (Sect. 3.1), fatigue load comparison (Sect. 3.2), and fatigue reliability comparison (Sect. 3.3). The reliability assessment also includes the sensitivity of the reliability level to different random inputs
at the target extended life (30 years) in all approaches. Finally, we present the overall discussions on the results in Sect. 3.4.

### 3.1   Comparison of turbulence levels

Figure 3 presents the comparison between site-specific turbulence based on met-mast anemometer measurements classified in different direction bins, the Frandsen model, and the IEC design class in different wind direction bins. The plot of each

direction bin only includes the mean wind speed bins in which there are enough available data to cover the comparison (more

than 20 points). The whisker box plots in Fig. 3 contain the data from 25% quantile up to 75% quantile, and the red "plus signs¨

account for data in the high and low tails and possible outliers. The upper and lower bounds represent 95 and 5 percentiles.

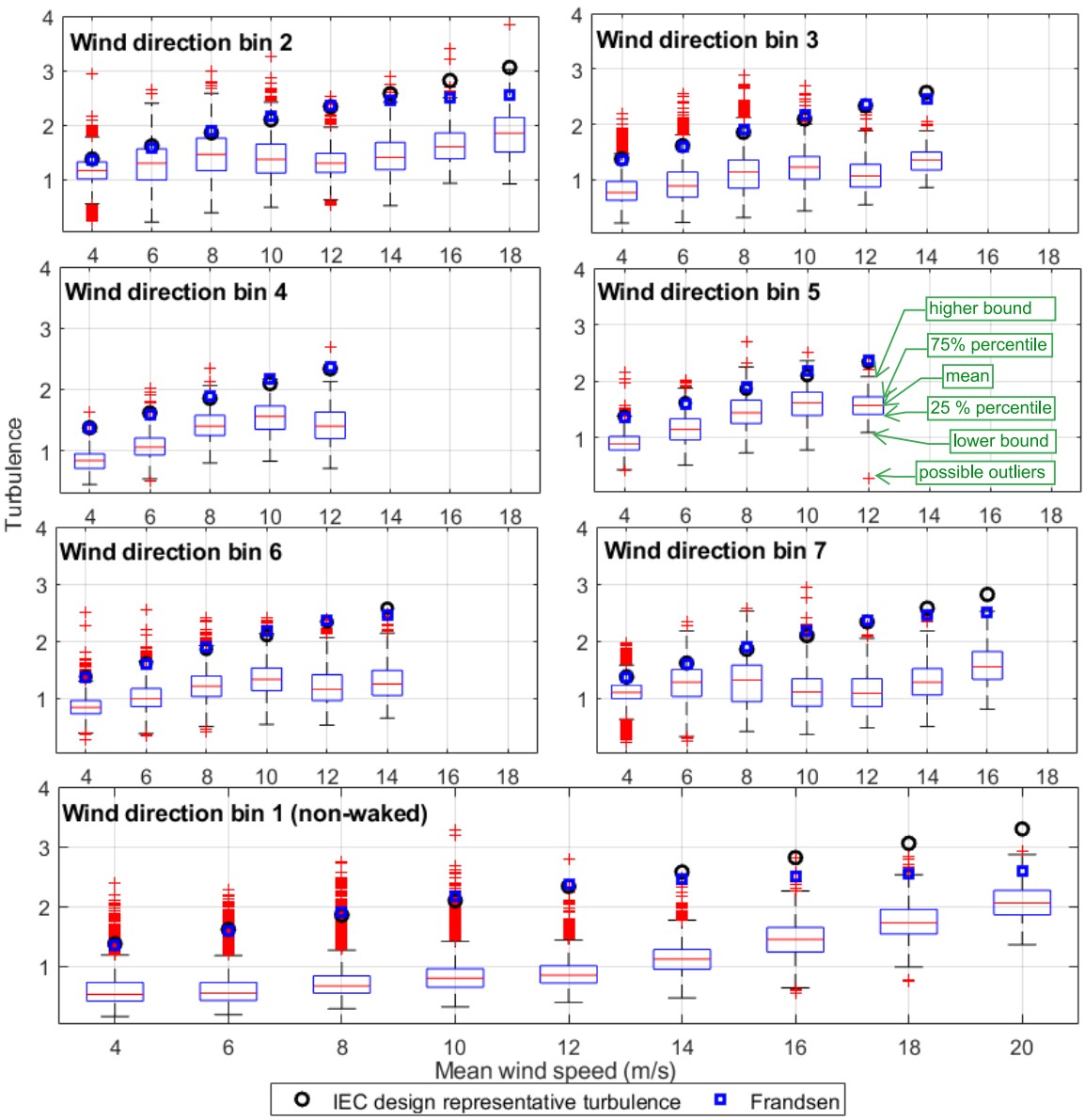

**Figure 3.** The enhanced turbulence (m/s) based on Frandsen estimation (blue squares) compared with IEC design representative turbulence
(black circles) and the site turbulence measurements (box plots)

The scatter of turbulence versus the corresponding 10-minute DEL measurements across different direction bins is provided in the Appendix (see Fig. C2 for a view of the high-turbulence points relative to the main data clusters). It should be noted that the location of the turbulence measurements differs slightly from the turbine location. Consequently, in some directions, greater deviations may occur between measured turbulence and what the turbine actually experiences due to specific wake effects. Such errors are negligible in free stream directions and higher in other directions. Assuming no outliers in the turbulence measurements and accepting the data as presented, Fig. 3 shows that in wind bin 1 (freestream conditions), the IEC design quantile underestimates the higher tail of site turbulence at low mean wind speeds, while overestimating it at high mean wind speeds (above rated speed). In addition, Frandsen model estimations are lower than design in high mean wind speeds while being the same as IEC representative value in low mean wind speeds. Emeis (2014) claims the same results for the case of design-level turbulence. This trend remains the same in almost all other wind direction bins. Direction bins 2 and 4 include very high wake effects and high turbulence levels and show less conservative assumptions by the IEC class and the Frandsen model even in high mean wind speeds. Fig. 3 reveals that in some cases, the Frandsen model does not produce much higher turbulence compared to the IEC design representative turbulence for the class. In the following section, we investigate the differences in terms of fatigue loads and fatigue reliability as more accurate parameters to estimate the lifetime extension based on each turbulence estimation approach.

## 3.2 $DEL_{lifetime}$ distributions

As shown in Eq. 14, finding the distribution of $log(DEL_{lifetime})$ is a necessity for estimating the probability of failure. The current section presents the distribution of $DEL_{lifetime}$ and $log(DEL_{lifetime})$ based on measurement data ($DEL_{10-min}$). First, we observe the empirical probability distribution of the 10-minute DEL measurements, from which we evaluate $DEL_{lifetime}$ realizations. Fig. 4 shows the empirical probability density of the 10-minute damage equivalent flapwise moment from measured in site.

As Fig. 4 reveals, the distribution of the 10-minute DEL data is multimodal and cannot be represented fully by unimodal distributions. The multimodality of the DEL can be a result of having both stratified and unstratified winds during the observation time, as the same behavior can be seen in each wind direction bin (see Fig. C3 in Appendix). We investigate the mixture of two or three gamma distributions as well as a mixture of two or three Gaussian distributions, as multimodal distributions have shown good candidacy for modeling of fatigue loads (see Mozafari et al. (2023a)). Among all, the mixture of two gamma distributions appears to be the best fit to describe the statistical behavior of the 10-minute damage equivalent flapwise moments in the case study wind turbine's blade root in all direction bins combined. Figure 5 shows this fit on the empirical CDF of data combined.

As Fig. 5 presents, the gamma mixture model fits the data well with fair accuracy in the higher tail. We use the probability of exceedance from this model and follow the steps presented in Sect. 2.4.3, to find the return load corresponding to 30 years. We use Eq. (10) and Eq. (11) to extrapolate the distribution to the load with 30 years return period. The slow growth of the tail from year 5 (corresponding to the probability of the largest data point observed) to 30 years shows that the distribution is fairly

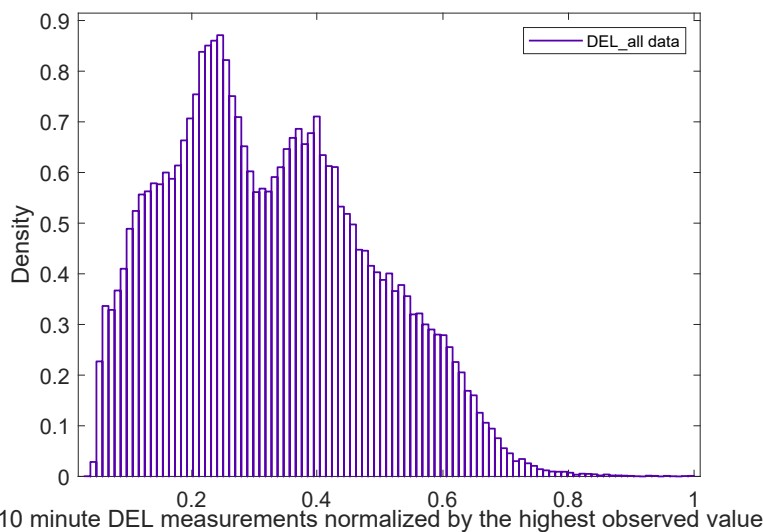

**Figure 4.** Empirical probability density of the 10-minute DEL measurements in all directions combined

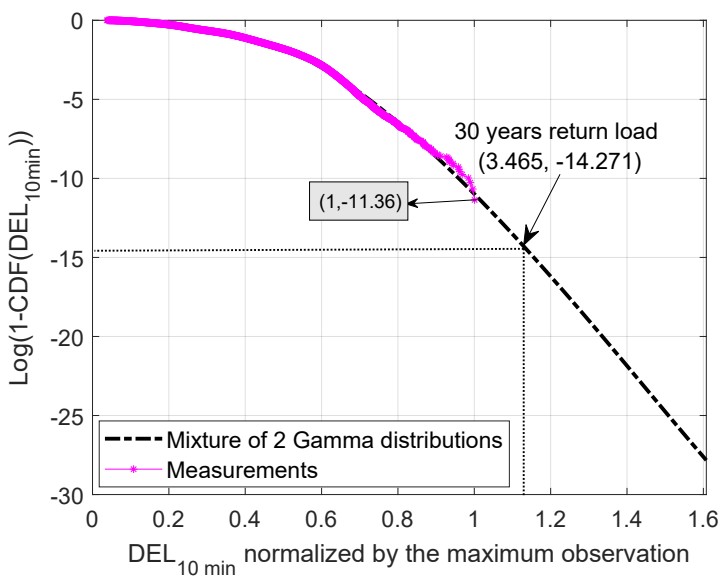

**Figure 5.** Probability of the exceedance of the DEL based on empirical cumulative distribution function (purple stars) and based on best distribution fit to the data (black dashed line)

converged, and a 5-year return period is enough for the main assumption in Sect. 2.4.3 to hold. Continuing the procedure with bootstrapping as outlined in section 2.4.3, the realization of $DEL_{lifetime}$ in different years is shown in Fig. 6.

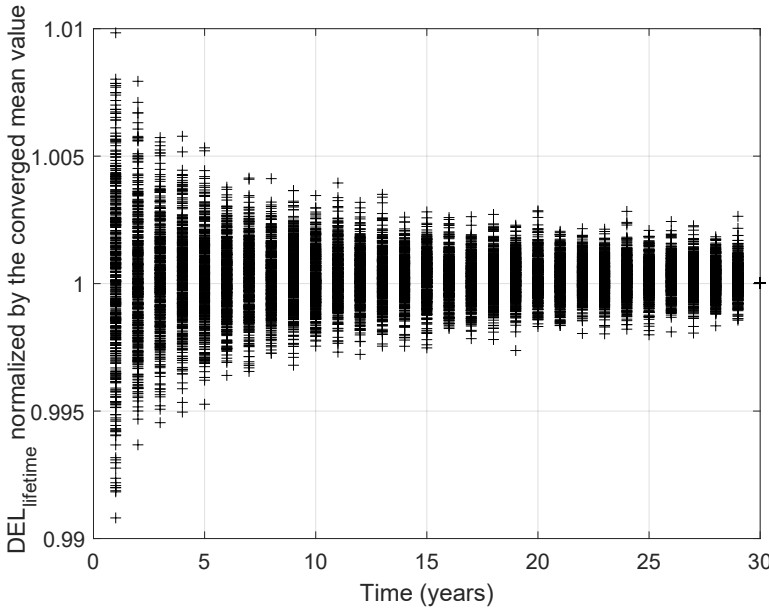

**Figure 6.** The realizations of $DEL_{lifetime}$ generated via bootstrapping of available 10-minute DEL estimates derived from measurement data

    As shown in Fig. 6, the mean value of the realizations converges as their standard deviation decreases. This complies with our expectation according to the law of large numbers: The mean converges to the "true" expected value of $DEL_{lifetime}$ as we gather more observations through the years. The change in the standard deviation with a slight change in the mean value

allows for the assumption of nonlinear damage accumulation through time (variable DEL through time). We use the converged distribution of $DEL_{lifetime}$ for the estimation of the annual reliability index.

    Figure 7 presents the probability density function (PDF) of $DEL_{lifetime}$ in two different turbulence scenarios of the IEC 61400-1 representative design value and the Frandsen estimation using bootstrapped data among simulations. The uncertainty of $log(DEL_{lifetime})$ in the site is modeled by a frequentist approach (maximum likelihood ) based on observations and

includes all sources of uncertainty. However, in the case of the other two approaches, the uncertainty of this parameter is assessed based on bootstrapping, and thus it only includes epistemic uncertainty. The data in Fig. 7 are normalized by the converged mean of $DEL_{lifetime}$ obtained above using gathered data.

    As Fig. 7 reveals, the estimation of the $DEL_{lifetime}$ based on the representative turbulence for design and the Frandsen estimation are conservative compared to the site-specific assessment based on measurements of turbulence. The Frandsen

model, in this case, leads to respectively less conservative fatigue loads than design-based approach as expected. In addition, the DEL realizations based on the Frandsen model are more spread, showing a higher variability. In the following section, we use the $DEL_{lifetime}$ distribution for assessing the fatigue reliability at the end of the design service life and the possible extended life in above-mentioned scenarios.

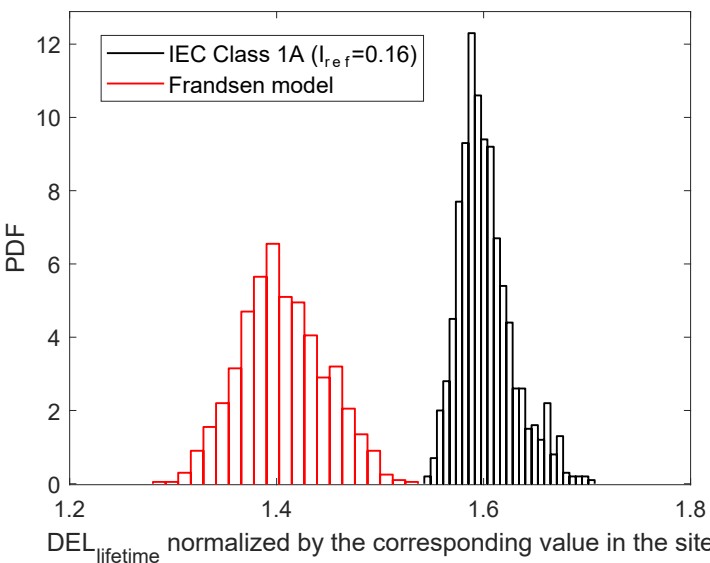

**Figure 7.** Probability density function (PDF) of $DEL_{lifetime}$ in two different scenarios of the IEC standard for design and site-suitability check (based on the Frandsen model) normalized by the mean $DEL_{lifetime}$ estimation based on site's measurements

As Fig. **??** shows, despite good coverage of the sample points among the scatter of wind measurements, the resulting 10-
430 minute DEL values are covering a lower bound of the load measurements at site. The errors mostly stem from the anemometer measurements. As a consequence, there is error in lifetime DEL. However, we use the data for estimating the reliability in the next section to illustrate the possible errors such data can bring to the assessment of lifetime extension.

### 3.3 Reliability and importance ranks

As the turbine model in HAWC2 simulations is generic and the material properties are not defined accurately, we assess the
435 lifetime extension after calibrating the material's mean strength. The calibration is made such that we get the target annual reliability index level for a moderate consequence of the failure of a structural component (equal to 3.3) IEC 61400-1 (2019) at the end of design life (25 years).
Fig. 8 presents the annual reliability index for 35 years in all case scenarios based on the calibrated material properties (mean value of strength (K) is multiplied by a factor).
The difference in the annual reliability levels in different scenarios in Fig. 8 is due to the difference in the turbulence, observed in Sect. 3.1, strain gauge calibration, possible missed controlling strategies for the neighboring turbines, as well as generic model errors. The reliability of 3.7 at year 35 for the Frandsen model means there is a possibility of a lifetime extension of up to 10 years for the turbine under study when using this model. Using the site data, this level is reached in more years, showing the conservative results from the Frandsen model. The steep declination in the design-based annual reliability curve in

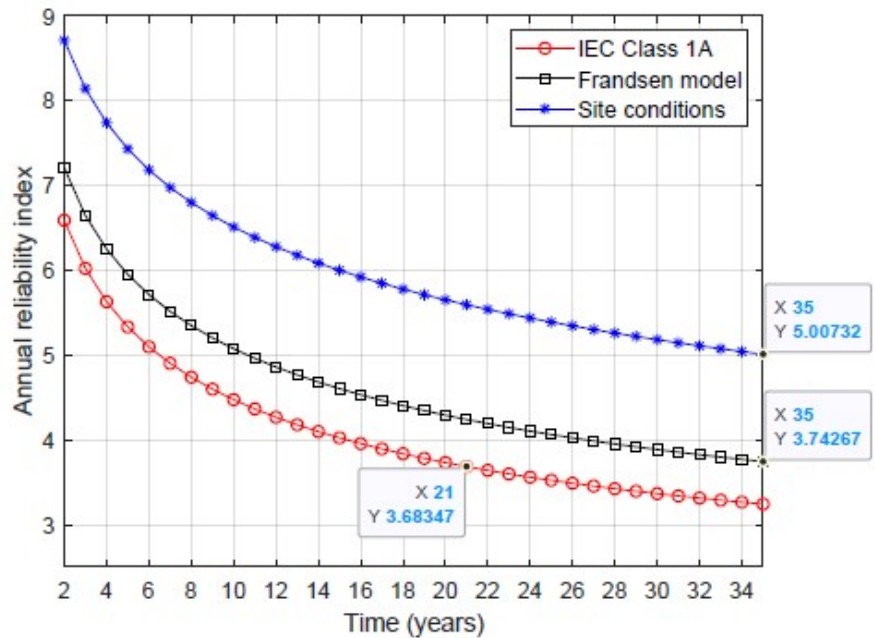

**Figure 8.** Annual reliability of the case study wind turbine in scenario 1: using Frandsen model (black square) and scenario 2: site load measurements (blue star) versus the IEC design class (red circle)

Fig. 8 represents the high mean value of the design DEL compared to the Frandsen model and site. The big difference between the reliability in the two scenarios is due to the high fatigue exponent of the composite increasing the effect of the mean of fatigue loads on the reliability.

Table 4 shows the importance rank of the random variables in the limit state function (see Eq. 14) based on FORM analysis.

**Table 4.** The sensitivity of the reliability to different random variables (%) based on scenarios 1 and 3 approaches versus design at year 30

| Random variables / Assessment basis | Frandsen model | IEC design-level | Site load measurements |
|---|---|---|---|
| $log(\Delta)$ | 38.29 | 41.70 | 44.46 |
| $log(K)$ | 47.84 | 52.09 | 55.54 |
| $log(DEL_{lifetime})$ | 13.87 | 6.21 | $7.62 \times 10^{-3}$ |

As Table 4 reveals, the relative importance of the load is low in all three scenarios. The relative importance of the fatigue load 450 is higher in the case of the Frandsen model because of the high coefficient of variation of the DEL in this case. Its sensitivity to loads is almost zero when it comes to the site-specific lifetime DEL. The effects of the uncertainty in the material properties are the highest in all cases because of the very high coefficient of variation. In the site assessment, the reliability is highly sensitive firstly to the material uncertainty and secondly to the damage accumulation rule.

## 3.4 Discussion

The data show the conservative estimations of turbulence based on the Frandsen model in high mean wind speeds and, thus, in the overall DEL estimations when considering the blade's flapwise bending moments. The reliability assessments show the possibility of a lifetime extension for more than 35 years while maintaining safety margins when using Frandsen method and a generic model

The difference between results of the two approaches (Frandsen model and load measurements) stems not only from differences in turbulence inputs, but also from possible generic model errors and strain gauge calibration errors. In this study, the generic model differs from the original only in controller settings, not in structural parameters. Moreover, according to Robertson et. al. (2019), the model parameters most relevant to fatigue loads are yaw angle error and certain structural parameters. Therefore, we do not expect generic model uncertainty to account for majority of the observed error.

The very low sensitivity of the reliability to the fatigue loads in the case of site-specific assessment (using loads measurements) is due to the relatively low coefficient of variation of this variable compared to material strength and Miner's rule. This shows the robustness of $DEL_{lifetime}$ as an accumulated/averaged random variable. The results comply well with the results shown in Mozafari et al. (2023b) showing how the accumulation decreases the coefficient of variation of $DEL_{lifetime}$.

There are some limitations and simplifications in the present study that must be considered and improved in future work. As an example, there is a potential difference in the results of the simulation-based approaches if the model was not generic and the simulations were offshore (considering wave loads). The results of the model validation (presented in the appendix) show that the small differences in load can lead to underestimation of the $DEL_{lifetime}$ in simulation-based scenarios and thus the
remaining service life. This is especially important in the assessment based on Frandsen model and generic model and as the sensitivity studies show that the reliability of the Frandsen-based approach is sensitive to the loads.

In addition, the results of the current study are performed on the blade flapwise load channel as a case study. However, all the load channels shall be investigated for lifetime extension estimation. Specifically, load channels with deterministic behavior (for example blade edgewise moments) often have lower design margins and thus are the critical components deriving
the lifetime extension of the wind turbine. The current study is not focusing on actual lifetime of the turbine and focuses on showing the difference between Frandsen site-suitability assessment and assessments based on site load measurements when it comes to fatigue reliability. Such comparison is more clear on a turbulence-driven fatigue load like flapwsie bending moment.

Furthermore, It has to be considered that the data with a return period of 5 years are only referring to the winter season and thus the tail shape of the distribution might be different if seasonal variability is also included and the data are collected

continuously.

Finally, although the turbulence levels from site and the Frandsen estimation are directly compared, the fatigue load results
based on the two are derived differently. The former is based on post-processing of strain gauge data and the later based on
aeroelastic simulations. Thus, the possible bias and errors of the turbine model and aeroelastic simulations can affect the DEL
and reliability comparison results.

For future work, we recommend the following studies:

1. Performing the same study on the performance of the Frandsen model in deeper locations within the wind farm, as they
   usually include more intense/complicated wake conditions.

2. Comparing use of ambient wind data (using Frandsen) with location-specific measurements of the turbulence considering
   all the load channels (lifetime extension assessment as a whole) and performing offshore simulations.

3. Considering other sources of uncertainty in the reliability assessment framework, including the uncertainty due to the
   range counting methods, uncertainty in simulations, etc.

4. Investigating the reasons behind the multimodality of the 10-minute DEL distribution.

5. Performing the same study using independent fittings and extrapolations of DEL in each wind bin for higher certainty
   and accuracy in the assessment.

6. Although Robertson et. al. (2019) conducted a sensitivity study on how aeroelastic model parameters affect fatigue
   loads, results for lifetime extension assessments may differ due to their comparative nature. For instance, parameter
   sensitivity under varying environmental conditions could influence outcomes. This matter is valuable for investigation
   in future work.

7. Joining the study with inspection and health-monitoring data coupled with risks and cost analysis to obtain a complete
   set of tools for decision-making regarding lifetime extension.

## 4  Conclusions

The objective of this study is to demonstrate the significant benefits of collecting and utilizing load/displacement measurements,
which can outweigh the challenges of such assessments by extending the project lifetime of wind turbines. The research mainly
compares two different data availability scenarios – one with structural response measurements and one without. In addition,
it addresses two common challenges in fatigue reliability assessments. First, it evaluates the performance of the Frandsen
model for estimating waked turbulence and corresponding fatigue loads at Lillgrund, a compact wind farm with mixed-wake

conditions. Second, it presents a methodology for extrapolating fatigue loads, showcasing an example application using strain gauge measurements in Lillgrund.

The results indicate that the Frandsen model can yield conservative fatigue load estimations at the investigated location despite the compact layout of Lillgrund.

Additionally, the relative importance of load estimates in assessments based on the Frandsen model is greater than in those based on site load measurements. Due to the way fatigue reliability decreases very slowly over the late years of project lifetime, the reduction in uncertainty created by load measurements at the site can provide the knowledge basis for many additional years of turbine operation within the acceptable reliability target.

The extrapolation approach presented in the current research facilitates the use of data when strain gauge measurements are unavailable for part or all of the turbine's lifespan. The assessment of the Frandsen model in this case study, representing a wind farm with short spacing, contributes valuable insights to ongoing research on the performance of this model in intense and mixed-wake conditions. Moreover, the findings on the robustness of reliability based on the load estimation approach are crucial for incorporating uncertainty into lifetime extension assessments.

Above all, the comparison between scenarios with and without load measurement and accurate measurements show the importance of having accurate environmental and load coupled with accurate models updating in real time with load measurements (digital twins).

## Appendix

## 1 Model Validation

As mentioned in Sect. 2.3.2, simulations in free-stream direction are used for validation of the model. The input is based on site-specific inputs from the non-waked freestream (wind direction bin 1). First, we compare the mean load levels from site measurements in wind bin 1 to the results of the group 3 simulation. Then, we compare the 10-minute DEL evaluations and investigate the differences in $DEL_{lifetime}$ formed via bootstrapping.

    Figures A1 and B1 show the comparison of the mean load levels and 10-minute DELs in the measurements versus simulations, respectively.

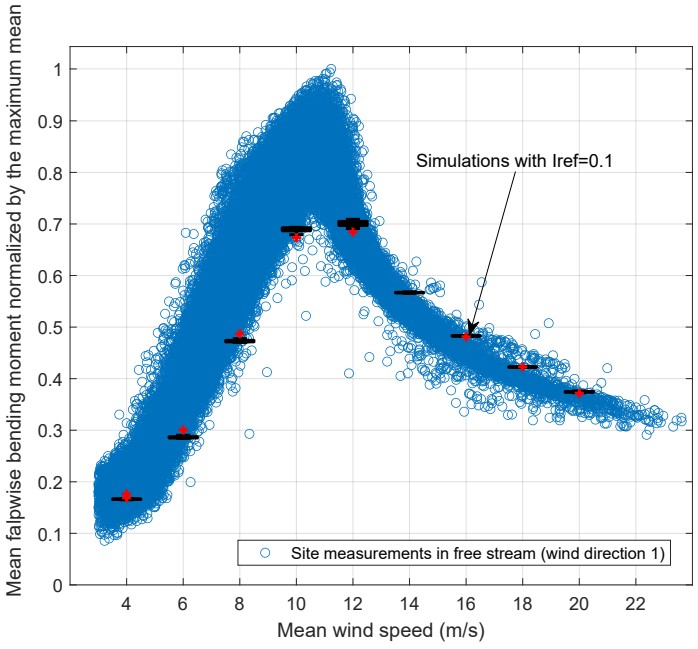

**Figure A1.** The mean flapwise bending moment in different mean wind speeds (measured by met-mast mounted anemometers) in non-waked directions from two sources of measurements (blue circles) and simulations (black boxplots)

    The data shown in Figure A1 and B1 reveal the high variation in the measured mean load and damage equivalent flapwise moment around rated mean wind speed. The high difference in this area can introduce some errors in the estimations based on simulations (groups 1 and 2) based on dominant (high probability) wind speeds. However, generally, the data show fair coverage of the site load behaviors.

Fig. C1 represents the power production versus mean wind speed compared with the nominal power curve after filtration.

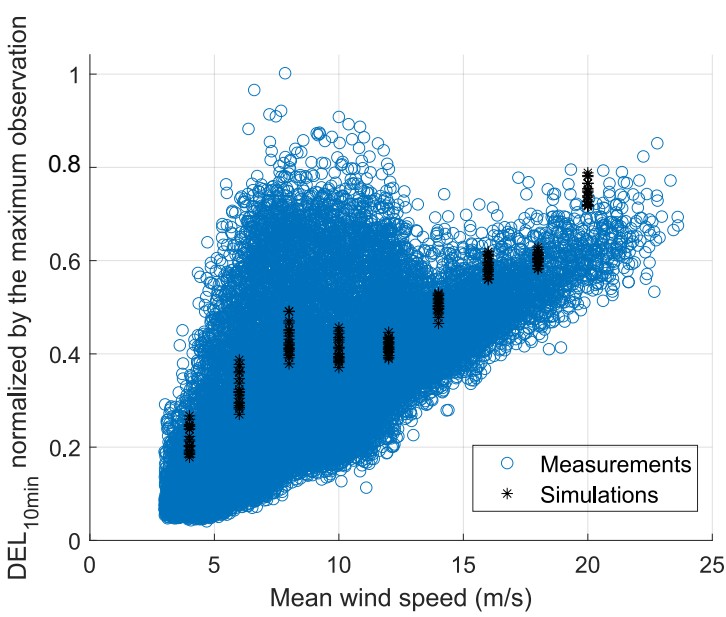

**Figure B1.** The cluster of 10-minute DEL data in the freestream versus the DEL estimations from simulations in each mean wind speed

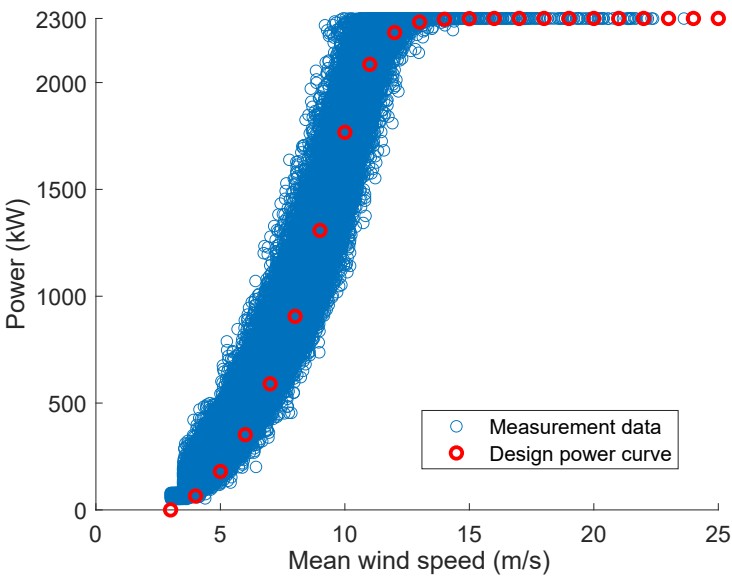

**Figure C1.** The filtered power production data versus mean wind speed, measured by met-mast mounted anemometer, (blue) compared with the nominal power curve (red)

Fig. C2 shows that the high tail of turbulence observations mostly belongs to the cluster of data, and the possibility of having a high number of outliers is low. The probability of exceedance of the 10-minute DEL measurements within each wind direc-

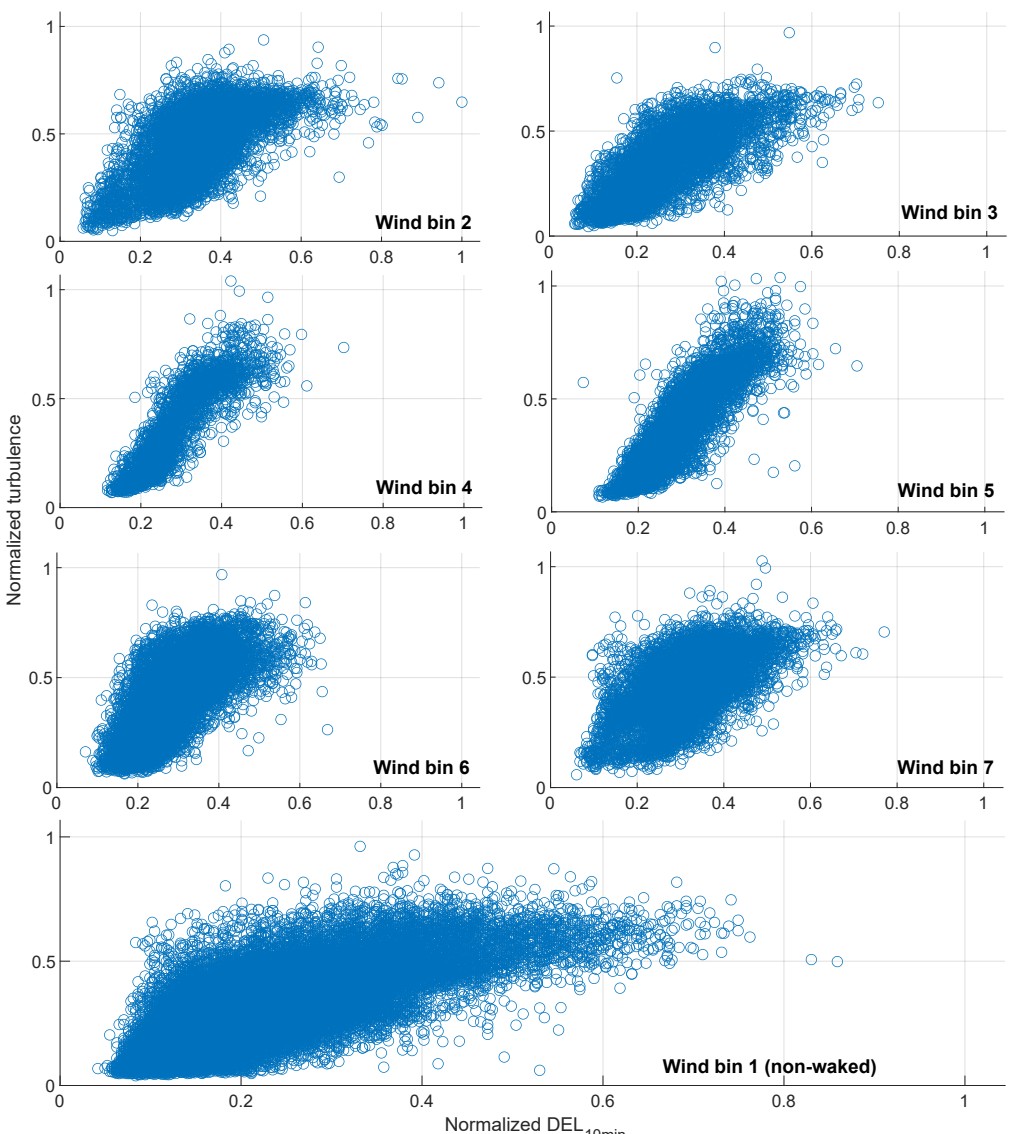

**Figure C2.** Scatter plot of turbulence data normalized by the highest observations versus corresponding 10-minute DEL observations at the same time in different wind direction bins before filtration

tion bin is shown in Fig. C3 using both the empirical CDF and the best distribution fit to each cluster of data.

Figure C3 shows that the highest DEL observations occur in wind bin 5.

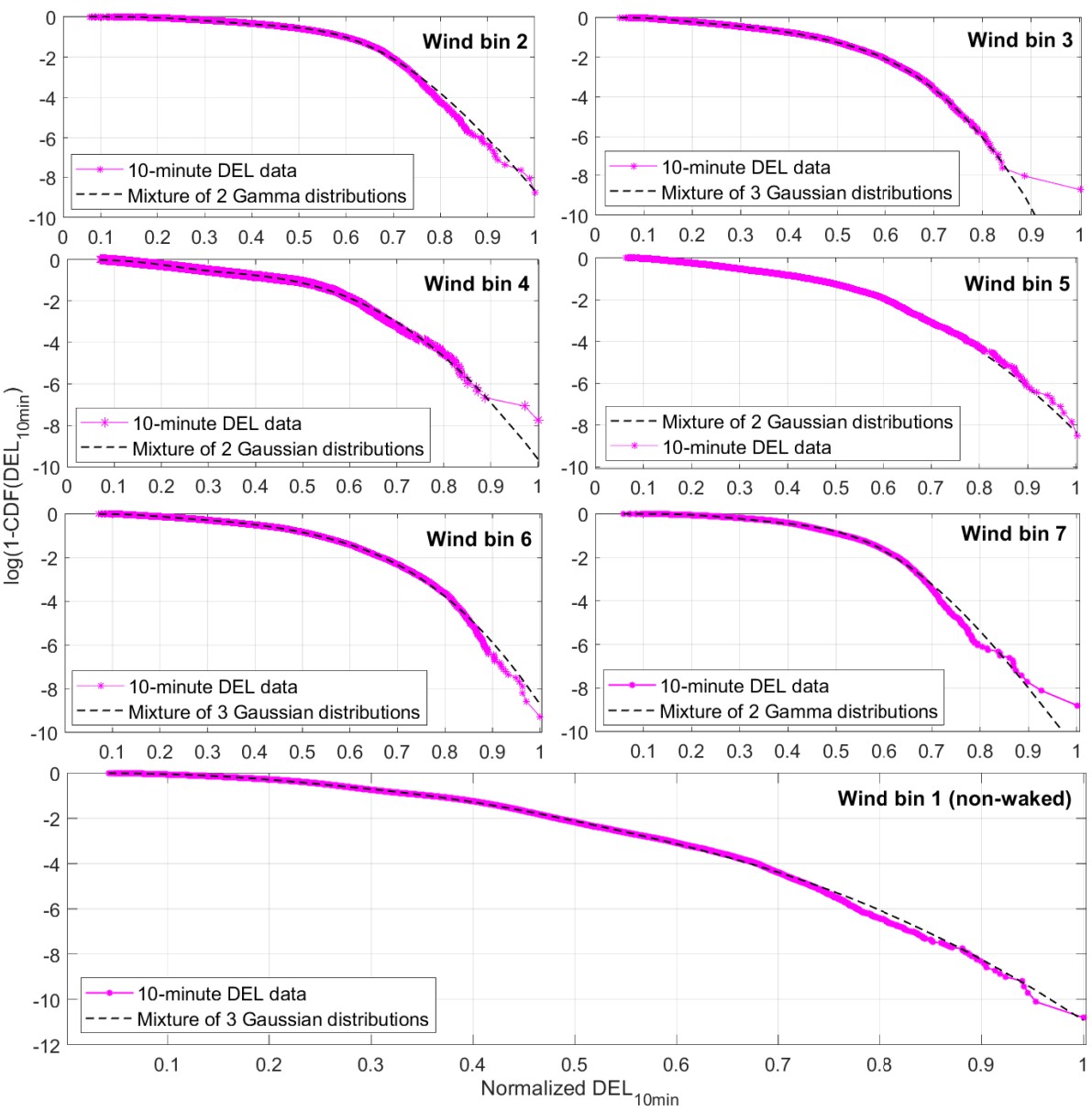

**Figure C3.** Logarithm of the probability of exceedance of the normalized $DEL_{10min}$ in different wind direction bins in the location of the wind turbine (purple dotted line) and the best distribution fits (black dashed-line)

In addition, the data in Table A1 reveal that the load fluctuations in the simulations are very small compared to reality. This is partly because of the integration over turbulence distribution (see Mozafari et al. (2024) for details) and partly due to the variability of the environmental condition (see Sect. 3.1). Although the validations show overestimations of the load and DEL in high mean wind speeds, we proceed with the study using the available HAWC2 model because the difference in the overall 555 DELs shown in Table A1 are less.

**Table A1.** Statistical parameters of $DEL_{lifetime}$ according to the non-waked aeroelastic simulations using site-specific turbulence and shear exponent versus the same parameters from measurements

| Source | Mean DEL normalized by the mean based on measurements | std of the $DEL_{lifetime}$ |
|---|---|---|
| Site-specific simulations for non-waked area | 1 | 0.014 |
| Measurements for non-waked area | 1.04 | 0.031 |

Different distributions shown in the table below are fitted to the mean wind speed data of the wind turbine. As can be seen, the best fit is the Rayleigh distribution. The maximum likelihood method is used for fitting and the prediction error is measured by Akaike information criterion (AIC).

**Table B1.** Different distributions fitted using maximum likelihood method to the wind speed measurements, including statistical parameters of the fits (shown as Par1, Par2, and Par3)

| Distribution | Par1 | Par2 | Par3 | Log-Likelihood | AIC |
|---|---|---|---|---|---|
| Rayleigh | 5.90 | | | $-7.23 \times 10^5$ | $1.45 \times 10^6$ |
| Gev | -0.12 | 3.70 | 5.47 | $-7.25 \times 10^5$ | $1.45 \times 10^6$ |
| Normal | 7.23 | 4.17 | | $-7.32 \times 10^5$ | $1.46 \times 10^6$ |
| gamma | 3.00 | 2.41 | | $-7.40 \times 10^5$ | $11.48 \times 10^6$ |
| Exponential | 7.23 | | | $-7.65 \times 10^5$ | $1.53 \times 10^6$ |
| Ev | 9.39 | 4.56 | | $-7.68 \times 10^5$ | $1.54 \times 10^6$ |
| Uniform | 0.00 | 2.81 | | $-8.75 \times 10^5$ | $1.71 \times 10^6$ |

**Table C1.** The best distribution fits to the $log(DEL_{lifetime})$ in case scenarios of Frandsen turbulence and IEC standard's representative turbulence for freestream

| Case | distribution of $log(DEL_{lifetime})$ | parameter 1 | parameter 2 | parameter 3 |
|---|---|---|---|---|
| IEC design based model | Normal | 0.2878 | 0.0143 | |
| Frandsen based model | Gev | -0.2536 | 0.0298 | 0.3772 |

*Author contributions.* SM, JR, and PV were responsible for the overall conceptualization of the study. SM wrote all the computer codes and performed all the data analysis. SM, PV, and KD were involved in the writing and editing of the manuscript

*Competing interests.* At least one of the (co-)authors is a member of the editorial board of Wind Energy Science.

Code and data availability

*Acknowledgements.* This work was authored in part by the National Renewable Energy Laboratory, operated by Alliance for Sustainable Energy, LLC, for the U.S. Department of Energy (DOE) under Contract No. DE-AC36-08GO28308. The views expressed in the article do not necessarily represent the views of the DOE or the U.S. Government. The U.S. Government retains and the publisher, by accepting the article for publication, acknowledges that the U.S. Government retains a nonexclusive, paid-up, irrevocable, worldwide license to publish or reproduce the published form of this work, or allow others to do so, for U.S. Government purposes. The work was mainly funded by and performed in DTU. We would like to thank DNV for support with providing the cloud facilities at a later stage, as well as Dr. Nikolay Dimitriv from DTU and Dr. Mahdi Teimouri Sichani from DNV for the consultations.

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
