# Peer review of "Added value of site load measurements in probabilistic lifetime extension: a Lillgrund case study"

_Wind Energy Science, 2024_

## Referee Comment (RC1)

**"Probabilistic lifetime extension assessment using mid-term data: Lillgrund wind farm case study" (Manuscript number: wes-2024-68)**

In this work, a probabilistic lifetime assessment of a wind turbine rotor blade is conducted. Three different approaches to determine the turbulence are compared: a standard IEC approach, the Frandsen model and using real measurement data. Furthermore, comparisons of the simulation results with real strain gauge measurements are done.

Fatigue assessments are an important topic in the context of wind turbines. Conducting them probabilistically is not yet state of the art and an important research topic. Nonetheless, in its current form, the manuscript is not sufficiently structured, explanations are missing and it features some mistakes. Hence, without a major revision, it is not suitable for a publication in the WES journal.

Comments:

1) The structure of the paper must be improved to make clear what the main innovation/topic is. Currently, it seems to be a mixture of "probabilistic fatigue assessment", "validation using real data" and "turbulence modelling".
    a. If the main topic is the probabilistic fatigue assessment, what is the difference between this paper and Mozafari et al. (2023) "Sensitivity…"
    b. If the main topic is the validation using in-situ data, more information regarding the measurement data must be given. Furthermore, in this case, a clearer focus on the results based on measurements and less work on simulations would be needed.
    c. If the main topic is the turbulence modelling and its effect on the turbine reliability (I think that this is the idea), the title, abstract and introduction must state this clearly.
2) Abstract: It remains unclear what the topic paper is (see comment 1)
3) In my opinion, the title of the paper does not represent in main topic of the work. Perhaps, turbulence modelling can be included in the title.
4) Introduction: The connection between the assessment using the Frandsen model (simulation-based, l. 24-44) and the limited data (measurement-based; l. 46-50) is unclear.
5) The state of the art (L. 52-67) is not sufficient and does not clearly differentiate between simulation-based and measurement-based approaches.
6) L. 121: Where exactly is the met mast situated? Please, show it in Figure 1.
7) L. 121: Are shadow effects of the met mast considered, e.g., reduced wind speeds if the anemometer lies behind the met mast.
8) L. 121: At which height(s) is the wind speed measured?
9) L. 124: Your data is biased, as you only cover periods in the winter/spring. This should at least be discussed. Is this bias relevant for your work?
10) L. 131: How much data has been removed?
11) Table 1: It is not clear for which time period the wind direction bin probabilities are given. Are these the probabilities for the same five years? And are they used somewhere. If yes, please highlight it. If not, you might just remove them.
12) Section 2.3.2: Your measurements come from an offshore turbine. The simulations seem to be done for an onshore turbine or all details regarding the offshore part are missing. Just simulating an onshore turbine and comparing it to offshore measurements does not seem to be sensible, even if you focus on blade loads.
13) L. 159: The site-specific turbulence distribution is not given, but only the reference turbulence intensity.
14) L. 162: How has the exponent of 0.1 been determined using in-situ measurement data?
15) Table 2: Why are the cut-in, the rated and the cut-out wind speed different compared to the real turbine (Section 2.1)?

16) L. 174: For groups 1 and 2 you use Rayleigh distributions (covering wind data of full years) whereas the biased measurement data (see comment 9) is used for the strain gauge-based approach. Hence, a direct comparison, as in Figure 7 is not possible.

17) Eq. (3) and (4) are not sufficiently explained, e.g., $di(\theta)$

18) Section 2.4.2: Formatting and explanations are not sufficient, e.g., $I_y$ and not $Iy$, $N_s$ is not explained etc.

19) Eq. (8) and (9): At the left side of the equation, the expectation E has to be removed, as $DEL^m_{lifetime} = E(DEL^m_{10min})$ and not $E\left(DEL^m_{lifetime}\right) = E(DEL^m_{10min})$

20) Eq. (9): Index i is missing.

21) L. 240 and l. 247-264: For me, it is not clear, why we need all this. If I understand it correctly, you fit a distribution to the 10min values (step 1). Then, you sample from this distribution to determine the lifetime value (step 3 and 4). Why do we need the DELs with long return periods. A single DEL with a high return period does not influence the overall lifetime DEL. Hence, they are not relevant and actually not used for the reliability assessment in Section 2.4.4.

22) L. 245: You neither show the fitted distribution for the lifetime DEL nor you state what type of distribution it is.

23) Eq. (10) where does this equation come from? It does not exactly match with Eq. (12), which is frequently used in literature.

24) Eq. (11): This equation is wrong, as it gives negative probabilities, since the CDF is always between 0 and 1.

25) Eq. (14) to (17): Please, revise these equations, as they are not always correct, formatting has to be improved and explanations are missing, e.g., $\Delta t$ and $P_f$ are not explained, it has to be $I_y$ and not $I$, the left side of Eq. (16) has to be $\Delta P_f(X, t + \Delta t)$, $m$ not $R$ etc.

26) L. 289: Why do you apply FORM and not MCS? Your limit state function can be evaluated computationally efficiently, so that MCS should not be a problem and MCS is more accurate.

27) L. 308: How do you define "enough data"?

28) L. 313: You state that the Frandsen model and the ICE design underestimate the turbulence for low wind speeds and overestimate it for high wind speeds. I cannot see this in Figure 14, e.g., the Frandsen model is above the 75% quantile for 4m/s and below the same quantile for 20 m/s.

29) L. 330: Why do you investigate this type of multi-modal distributions and not others?

30) L. 334-344 and Figure 4 and 5: Why do we need this? For Section 3.3, it is not needed.

31) L. 336: You state that "the probability of the largest data observed" corresponds to five years. However, this is not correct, since you do not have data of five full years.

32) Table 4: How did you determine the sensitivities?

33) Table D1: How are the parameters of the different distributions defined?

Typos etc.:

34) L. 69: "assess" not "assesses"

35) L. 86 and others: "Section" and not "Sect." or "section". Same applies to "Eq.", "Table" etc. Or at least be consistent.

36) L. 133: "in Table 1" not "in 1"

37) L. 138: I think it is "D1" and not "D2". Overall, reference to figures in the appendix are not always correct.

38) L. 174: "Rayleigh" not "Reighley"

39) L. 241: $365 \times 24$ … not $365 * 24$ …

40) L. 346: "in Fig. 6" not "in 6"

41) Figure 6: $I_{ref}$ not Iref

42) Table 4 (and appendix): Do not use the notation 7.62e-3, but $7.62 \times 10^{-3}$

43) L. 392: "fatigue" not "Fatigue"

44) L. 419: "h and more"?

45) L. 446: "In the following sections, we compare the turbulence levels in three scenarios of the study"?
46) Caption of Table D1 has to be corrected.
47) Caption of Figure D2 has to be corrected.

---

## Author Comment (AC1)

**Response to referee #1 (Manuscript number: wes-2024-68):**

*We appreciate the detailed comments from the reviewer as we believe implementing and addressing them have improved our paper to a high extent.*

*The reviewer notes and comments are presented in black, and our corresponding responses are presented in blue.*

"Probabilistic lifetime extension assessment using mid-term data: Lillgrund wind farm case study" (Manuscript number: wes-2024-68)

In this work, a probabilistic lifetime assessment of a wind turbine rotor blade is conducted. Three different approaches to determine the turbulence are compared: a standard IEC approach, the Frandsen model and using real measurement data. Furthermore, comparisons of the simulation results with real strain gauge measurements are done. Fatigue assessments are an important topic in the context of wind turbines. Conducting them probabilistically is not yet state of the art and an important research topic. Nonetheless, in its current form, the manuscript is not sufficiently structured, explanations are missing, and it features some mistakes. Hence, without a major revision, it is not suitable for a publication in the WES journal. Comments:

1) The structure of the paper must be improved to make clear what the main innovation/topic is. Currently, it seems to be a mixture of "probabilistic fatigue assessment", "validation using real data" and "turbulence modelling". a. If the main topic is the probabilistic fatigue assessment, what is the difference between this paper and Mozafari et al. (2023) "Sensitivity…" b. If the main topic is the validation using in-situ data, more information regarding the measurement data must be given. Furthermore, in this case, a clearer focus on the results based on measurements and less work on simulations would be needed. c. If the main topic is the turbulence modelling and its effect on the turbine reliability (I think that this is the idea), the title, abstract and introduction must state this clearly.

Thank you for sharing your thoughts which shows that other readers may also face unclarities based on the structure of the pre-print. Thus, we have updated the abstract and introduction to make it more clear for the reader and hopefully answer all the above questions. Below explanation is the response to comment #1 and it is also added to the introduction in its current form:

*'When it comes to lifetime extension of wind turbines in a wind farm, one must re assess the service lifetime by replacing the design assumptions with the conditions experienced in the site. In such re-assessments, normally, fatigue is the main subject of interest because of its direct functionality of time. The information about lifetime in site can be gathered in different manners based on data availability. Some of the common scenarios are as below:*

1. *In case only free-stream turbulence measurement is available, one can estimate the waked turbulence in each turbine's location using simplified models like Frandsen (Frandsen, 2007; Frandsen and Madsen, 2003), suggested by IEC 61400-1 (2019) for site suitability checks. The corresponding estimations are then used to perform aero-elastic simulations. Site-specific lifetime can be estimated using the resulting fatigue loads.*

2. *The turbulence measurements in the turbine's specific location might be available. In such scenario, one can use the measurements as inputs to the aeroelastic simulations and perform fatigue assessments. The time at which fatigue reliability reaches the target level can then be derived.*

3. *In some cases, the structural response (load/displacement) measurements are available for a limited duration of the lifetime in a specific hotspot. In case Supervisory Control and Data Acquisition (SCADA) also exist, one can form a digital twin for deriving the loads in other components/locations. On the other hand, direct utilization of the measurement data for assessing lifetime extension is also an option. However, the later involves challenges like spatial and temporal extrapolations.*

*Often the structural response measurements in the site are owned by the turbine manufacturer and are not accessible for the wind farm owner/developer. In addition, the measurements are not gathered for a long time or in many locations. The purpose of the current research is to showcase the differences of lifetime extension assessment in different scenarios (with and without load/displacement measurements) using a case study wind turbine for which all the above-mentioned scenarios are feasible. Additionally, the study tackles two common challenges in scenarios with and without structural response measurements. First, it addresses the question of performance of the Frandsen model—as a simplified approach for estimating enhanced turbulence due to wakes— in a compact wind farm layout. Second, we present a method for statistical extrapolation of mid-term strain gauge measurements for estimating long-term fatigue loads.'*

The rest of the introduction together with abstract and conclusion are also edited and modified accordingly to make the purpose and outcomes clearer.

2) Abstract: It remains unclear what the topic paper is (see comment 1)

Abstract is updated now to clarify the main intention of the research and outcomes.

3) In my opinion, the title of the paper does not represent in main topic of the work. Perhaps, turbulence modelling can be included in the title.

Thank you for sharing your thoughts. The title is now changed to '*Added value of site load measurements in probabilistic lifetime extension: a Lillgrund case study*' to better represent the main purpose of the paper (see response to comment #1).

The waked turbulence estimation is not the main purpose and is one of the two additional results (as mentioned in the new introduction). Thus, we keep it outside of title to prevent possible confusions.

4) Introduction: The connection between the assessment using the Frandsen model (simulation-based, l. 24- 44) and the limited data (measurement-based; l. 46-50) is unclear.

The whole introduction modified now to better represent the bigger picture and the full purpose of the paper with connecting different pieces.

5) The state of the art (L. 52-67) is not sufficient and does not clearly differentiate between simulation-based and measurement-based approaches.

Introduction rephrased and updated now.

6) L. 121: Where exactly is the met mast situated? Please, show it in Figure 1.

Added in figure 1 now with reference in L. 124. Thanks for mentioning.

7) L. 121: Are shadow effects of the met mast considered, e.g., reduced wind speeds if the anemometer lies behind the met mast.

This is a very relevant point. Thank you for mentioning. The met mast used for measurements of wind speed is placed on a pole on the top of the tower and thus there is no shadow effect included. The information is now added to the text (lines 124 and 125 of the updated paper) for clarification with an additional reference to the report on meteorological conditions of Lillgrund which includes more details.

8) L. 121: At which height(s) is the wind speed measured?

65 meters- added to the text (in L. 125) now for clarification.

9) L. 124: Your data is biased, as you only cover periods in the winter/spring. This should at least be discussed. Is this bias relevant for your work?

If you are referring to figure B1, it is misleading as it shows DEL versus time while time is in a special format of 'yyyymmddtttt' (tttt being the time for ex: 1130 means 11:30). Although the data are not continuously measured for the full 5-year period, they cover months #10, 11, and 12 in 2008 and all months in 2009, 2010, 2011 and 2011. Thus, although the duration is not fully covering 5 years it is representative of all the seasonal variations. The figure is replaced by a text explaining the data for more clarity. The explanation added is as below (as reference):

'*The measurement campaign has been running in 5 years but not continuously. The data covers about two years in terms of duration length. It includes different timings in the last 3 months of 2008 and all the months in 2009, 2010, 2011 and 2012. Thus, some data with a return period of 5 years are included among measurements*.'

The last sentence of the explanation above is also a response to comment #31.

10) L. 131: How much data has been removed?

88031 data remained. How many data did we have is unclear and unfortunately, we do not have access to the data anymore. A line is added to the end of the paragraph:

'*A total of 88031 data points remain after the filtration*.'

11) Table 1: It is not clear for which time the wind direction bin probabilities are given. Are these the probabilities for the same five years? And are they used somewhere. If yes, please highlight it. If not, you might just remove them.

The table shows the number of data points in 5 years (added clarification in the text). It is presented to show how the available data can represent the probability of each wind speed bin presented in another work (referenced in the table).

Yes, the values of probability are used to weight the DELs of each bin.

12) Section 2.3.2: Your measurements come from an offshore turbine. The simulations seem to be done for an onshore turbine or all details regarding the offshore part are missing. Just simulating an onshore turbine and comparing it to offshore measurements does not seem to be sensible, even if you focus on blade loads.

Thank you for your relevant comment. The results of the load measurements on the channels show a good alignment with the measurements (Figure A1 for mean load values and table A1 for standard deviation of the load) and thus are reliable for the load channel under study. However, we agree that there is a weakness of the current work and must be emphasized more clearly in the discussions (unfortunately we have missed this important point in the current version). Explanation added now in the discussions (in point #2 in 'discussions').

13) L. 159: The site-specific turbulence distribution is not given, but only the reference turbulence intensity.

There is a mistake in the two columns which is now corrected. A description as below is also added to the ending of 2.3.1 for clarity:

'*In the current case, different distributions best describing the turbulence in each wind speed in the free stream is used. However, we do not present the details of those fits to be concise.*'

14) L. 162: How has the exponent of 0.1 been determined using in-situ measurement data?

It is not based on measurements. It is an estimation based on smooth terrain (open water) condition of the offshore wind farm. This description is added now to the text for clarity. A new reference for shear exponent estimation based on lidar measurements is also added ('Liew J, Göçmen T, Lio AW, Larsen GC. Extending the dynamic wake meandering model in HAWC2Farm: a comparison with field measurements at the Lillgrund wind farm. Wind Energy Science. 2023 Sep 8;8(9):1387-402.')

15) Table 2: Why are the cut-in, the rated and the cut-out wind speed different compared to the real turbine (Section 2.1)?

That was a mistake, and the table is corrected now.

16) L. 174: For groups 1 and 2 you use Rayleigh distributions (covering wind data of full years) whereas the biased measurement data (see comment 9) is used for the strain gauge-based approach. Hence, a direct comparison, as in Figure 7 is not possible.

The purpose of the current research is to illustrate different scenarios and show how different the results can look like for the wind farm developers in real scenarios. The current case study, the available measurements and the generic model in hand are all representative of the common case scenarios (in fact one of the best availability of data). The purpose is not to differentiate between different theories for a theoretical case but showcase real scenarios of assessment.

17) Eq. (3) and (4) are not sufficiently explained, e.g., $di(\theta)$

Thank you for your comment. A description of unknown parameters including necessary references is now added to the equation and a footnote: '*For further details and derivation of the equations 3 and 4, see (Frandsen, 2007) and (IEC 61400-1, 2019)*' is also added now.

18) Section 2.4.2: Formatting and explanations are not sufficient, e.g., $Iy$ and not $Iy$, $Ns$ is not explained etc.

Corrections on formatting applied and further explanations added now.

19) Eq. (8) and (9): At the left side of the equation, the expectation E has to be removed, as $DEL_{lifetime}\,m = E(DEL_{10min}\,m)$ and not $E(DEL_{lifetime}\,m) = E(DEL_{10min}\,m)$

Agreed (as the result would just be a realization of DEL_lifetime^m based on the number of DEL_10min realizations it may get close to the estimated value). Thus, both equations are modified and corrected now.

20) Eq. (9): Index i is missing.

It is relatively small; however, it is there. Parentheses are added to make it clearer.

21) L. 240 and l. 247-264: For me, it is not clear, why we need all this. If I understand it correctly, you fit a distribution to the 10min values (step 1). Then, you sample from this distribution to determine the lifetime value (step 3 and 4). Why do we need the DELs with long return periods. A single DEL with a high return period does not influence the overall lifetime DEL. Hence, they are not relevant and actually not used for the reliability assessment in Section 2.4.4.

The effect on the mean value will be small but not zero. The importance is discussed in the literature review in the introduction. However, for the sake of clarity, a reference to Mozafari et al. (2023a) and Mozafari et al. (2023b) is added -to show the necessity of such investigations- as below:

'(For reference to the importance of statistical extrapolation in estimation of DELlifetime please see (Mozafari et al. (2023a)) and (Mozafari et al. (2023b))'

22) L. 245: You neither show the fitted distribution for the lifetime DEL nor you state what type of distribution it is.

This line is a part of description of the general methodology. The distribution of the DEL based on the current data is shown and discussed later in results.

23) Eq. (10) where does this equation come from? It does not exactly match with Eq. (12), which is frequently used in literature.

Reference and extra explanations are now added (lines 280-286).

24) Eq. (11): This equation is wrong, as it gives negative probabilities, since the CDF is always between 0 and 1.

The 'Log' sign was extra and is now excluded- Thank you for the correction.

25) Eq. (14) to (17): Please, revise these equations, as they are not always correct, formatting has to be improved and explanations are missing, e.g., $\Delta t$ and Pf are not explained, it has to be Iy and not I, the left side of Eq. (16) has to be $\Delta P_f(X, t + \Delta t)$, $m$ not $R$ etc.

Revisions are made as below:

1. R is correct. Description added in parentheses.
2. Eq 6 is corrected now.
3. Formatting of eq. 4 is improved with addition cross signs
4. $\Delta t$ and Pf and other parameters in equations 16 and 17 are now explained.

26) L. 289: Why do you apply FORM and not MCS? Your limit state function can be evaluated computationally efficiently, so that MCS should not be a problem and MCS is more accurate.

*Reference to the reasons for choice and the comparison for a similar case is provided in (Mozafari et al. (2024))'*

27) L. 308: How do you define "enough data"?

*Text modified as: 'The plot of each direction bin only includes the mean wind speed bins in which there are enough available data to cover the comparison (more than 20 points)'. This choice is very qualitative, as some bins had very few data (even less than 10) because of low probability of occurrence.*

28) L. 313: You state that the Frandsen model and the ICE design underestimate the turbulence for low wind speeds and overestimate it for high wind speeds. I cannot see this in Figure 14, e.g., the Frandsen model is above the 75% quantile for 4m/s and below the same quantile for 20 m/s.

*We assume the reference is Figure 2. Agreed that the wording must be corrected. Below correction is made:*

*'If we consider no outlier in turbulence measurements and approve the data as they are, according to Fig. 2 in the wind bin 1 (free stream condition), the Frandsen model and the IEC design level turbulence underestimate the higher tail of the site turbulence in low mean wind speeds while overestimating it in high mean wind speeds (over the rated speed). In addition, Frandsen model estimations are higher than design in high mean wind speeds while being the same as IEC representative value in low mean wind speeds'*

29) L. 330: Why do you investigate this type of multi-modal distributions and not others?

*The text is modified to:' We investigate the mixture of two or three Gamma distributions as well as a mixture of two or three Gaussian distributions as multimodal distributions have shown good candidacy for modelling of fatigue loads (see (Mozafari et al., 2023a)'*

30) L. 334-344 and Figure 4 and 5: Why do we need this? For Section 3.3, it is not needed.

*This section is answering one of the side questions of the research showcasing the performance of the Frandsen model in different wake scenarios (different wind bins).*

31) L. 336: You state that "the probability of the largest data observed" corresponds to five years. However, this is not correct, since you do not have data of five full years.

*In fact, data includes points with return period of 5 years (Kindly refer to reply of comment #9). However, the consideration that the tail is representative for only one season shall be included. This is added now to the discussion section.*

32) Table 4: How did you determine the sensitivities?

*As mentioned in the table description, they are 'importance rank of the random variables'. Explanation and reference are added in the methodology now for more clarity (lines 312-313).*

33) Table D1: How are the parameters of the different distributions defined? Typos etc.:

As mentioned in the description (L. 515): 'The *maximum likelihood method is used for fitting and the prediction error is measured by Akaike information criterion (AIC).*' However, for more clarification, the table title is also modified to be more descriptive.

34) L. 69: "assess" not "assesses"

Corrected.

35) L. 86 and others: "Section" and not "Sect." or "section". Same applies to "Eq.", "Table" etc. Or at least be consistent.

Corrected the 'section' and checked the whole text for consistency and for aligning with WES journal guidelines (In the beginning of sentences 'Section / Figure 'and in the middle of the sentences 'Sect. / Fig.')

36) L. 133: "in Table 1" not "in 1"

Corrected. Thank you.

37) L. 138: I think it is "D1" and not "D2". Overall, reference to figures in the appendix are not always correct.

The appendix numbering and referencing is now updated. Thank you for mentioning.

38) L. 174: "Rayleigh" not "Reighley"

Thanks for noticing. Corrected now.

39) L. 241: $365 \times 24$ ... not $365 * 24$ ...

Applied.

40) L. 346: "in Fig. 6" not "in 6"

Corrected.

41) Figure 6: Iref not Iref

Corrected.

42) Table 4 (and appendix): Do not use the notation 7.62e-3, but $7.62 \times 10^{-3}$

Corrected now.

43) L. 392: "fatigue" not "Fatigue"

Corrected. Thanks for noticing.

44) L. 419: "h and more"?

Typo. Deleted now. Thank you.

45) L. 446: "In the following sections, we compare the turbulence levels in three scenarios of the study"?

Editing mistake. Omitted now. Thank you.

46) Caption of Table D1 has to be corrected.

Corrected.

47) Caption of Figure D2 has to be corrected

Corrected now according to comment #24 reply- additional information added to the caption as well.

---

## Author Comment (AC2)

We would like to truly thank the reviewer for the detailed and very relevant comments that helped us improve the work to a high extent.

*The reviewer notes and comments are presented in black, and the responses are presented in blue.*

Lifetime extension of wind turbines is a very important topic of high industrial relevance. Using a probabilistic approach is also very relevant. Therefore, the paper is of interest to be published. However, there are a number of unclear sections and missing explanations, see below. A major revision is recommended.

Detailed Comments:

Line 79

' One must consider that the material properties are calibrated such that the target reliability level of 3.7 ISO-2394 (2015) is reached': this is an annual reliability index? And why 3.7? which components are considered?

Here, only the blade is considered. Assuming a moderate consequence of failure, a target reliability index of 3.7 can be assumed based on ISO 2394. However, looking at your later comment, referring to Section 2.4.4, we agree that we must stay aligned with IEC 61400-1 basis for calibration of safety factors for the results of all probabilistic assessments to be comparable. We would like to thank you for your important comment. The modifications are added now to the text. Considerations of the offshore version (25years of design lifetime for the type) of SWT-3.2MW, which was missing previously, is also added.

Line 80

'the levels are not': unclear – reformulate

Reworded to 'magnitudes' to make it clearer.

Line 161

Which 'exponent'? wind shear?

Yes- changed to 'shear exponent' in the text for clarification.

Table 2

Is full (Weibull) distribution n of turbulence used as specified in IEC 61400-1:2019? And if not add a comment on the potential influence.

1:

No, only the 90% quantile is used as in Ed.1. This is a good point. Thank you for your suggestion. Explanation added now as below:

*'It shall be noted that group one is based on Edition 1 of the IEC standard and in case a full Lognormal or Weibull distribution are used (as in Editions 3 and 4, respectively), the results of the study differ (See (Mozafari et al., 2024) for differences). The following section includes the mathematical relations and procedures used in the study.'*

Line 174

Rayleigh

Corrected.  Thank you.

Line 174 + 178

Why use two different editions. Use ed 4 in order to obtain up-to-date comparisons?

The study shows the results based on Ed. 1 normal turbulence model (90% quantile as the representative value of turbulence in each wind speed bin). However, the difference that using Ed. 4 can make is shown in another research and the possible effect on the current study is discussed (see response to comment above regarding table 2).

Line 190

Explain equation

 Explanations added now with a reference to IEC 61400-1 for further details.

Eq (4)

Missing m in eq?

Corrected now and fortunately it was only a mistake in the text not in the procedure. Thank you.

Eq (6)

k?

Explanation for 'k' is added now.

M to power m?

Clarified with addition cross sign and parentheses.

for composites the mean stress level is important. How is that accounted for?

1:

The mean stress level correction is not applied in calculations of DELs. However, since the aim of the work is to compare different approaches of obtaining DEL (all without correction), the effect on the results is low.

Line 213-215

Unclear – reformulate

Edited as:

*'In Eq. (2), $\sigma$ is the turbulence standard deviation (turbulence) of the free stream wind (ambient flow) considered as a random variable. In addition, $\mu_\sigma$ and $\sigma_\sigma$ refer to the mean and standard deviation of the turbulence, respectively'*

Eq (8)

How is lifetime damage obtained from 10-min damage?

By taking the weighted mean (by probabilities) of values of DEL_10min^m. Here the estimated value of (DEL_lifetime) was a mistake and has been omitted from the left side of the equation.

Line 218

'Probability of turbulence': which turbulence (ambient, effective, …) is the probability linked to?

The word 'directional' is added for clarity.

Line 221

Conditional probabilities?

Yes! Reworded with 'conditional'. Thank you!

Line 235

Explain why 'log' is used

We refer to usage in equation 14 which is in form of log. A text is added.

Line 240

Describe what is 30-year return loads'. Is it 30-year extreme loads to account for the extreme loads being important due to the high Wohler exponent?

A reference is added to the relevant equations (and explanations) for return load in step 2.

Yes. The return load is used to extrapolate the tail of the distribution; description in the updated document:

1:

*'The extrapolation is used to complete the tail of the DEL10min distribution to account for highest values that might change the weighted mean value (DEL_lifetime) if included. These values can have high effect due to the high fatigue exponent of the composite ((Mozafari et al., 2023b))'*

Line 240

'Forming a database based on the distribution': unclear – explain which distribution. If the realizations follow the distribution function how is new information obtained?

Rewording is done now in step 2 with added reference to the corresponding explanations as below:

'2- Forming a database based on the distribution found in step 1 and extrapolating to 30-year return loads (Eq. 10 to 12)'

(previous reference in the next paragraph was a typo and now is corrected from 'Step 1' to 'Step 2')

Eq (10) +(11) +(12)

Explain the probabilistic assumption behind eq (10)

Explanation and reference added now as below as below:

*'Equation 10 is extracted from formula of probability of exceedance a threshold level (here the load which happens once every 30s) assuming a Poisson process for describing the peaks over threshold problem (for further information see (de Oliveira JT, 2013)). In the current case, the frequency of exceedance is $1/T_{LR}$'. It has to be noted that Eq. 10 is correct when $T_{LR}$ is relatively large (here, equal to the number of 10 minutes in 30 years).'*

Eq (1): Lr is not included in the right-hand side of the equation?

We assume you are referring to equation 10. A modification in Eq. 10 is made as follows:

$T \rightarrow T_{LR}$

In addition, explanations added as mentioned above describe how '$T_{LR}$' is related to the LR.

Probabilities in eq (11) always between 0 and 1?

Yes, the 'log' sign is now omitted to make the formula correct (only text mistake and not the applications).

Explain reference times for the probabilities

Explanations added as mentioned above.

Are the loads obtained 'random point in time' loads or maximum loads with a certain reference period?

First, the maximum loads with a certain reference return period are defined, and the frequency of lower loads is derived accordingly.

1:

Above statement is added to the text for clarification (line 290 in the updated document).

Section 2.4.4

The Frandsen and IEC turbulence models together with partial safety factors are intended for deterministic design and not for probabilistic design and reliability analysis. This link should be included in the probabilistic formulations.

We very much appreciate your relevant and helpful comment.
Below paragraph is added to the end of section 2.4.4:
*'It must be noted that the Frandsen and IEC turbulence models together with partial safety factors are intended for semi-deterministic design and not for probabilistic design and reliability analysis. However, since the partial safety factors are calibrated based on achieving certain reliability level (to which we are also setting the values for) at the end of the design lifetime, the results are comparable. Such comparisons are presented in the next section.'*

Line 268

'Probability of failure at time t and can be stated as the probability of exceeding a certain level': this probability is the probability of failure at time t and not the accumulated probability of failure up to time t and also not the annual probability of failure?

Thank you for noting. To avoid misleading, the paragraph is reworded as below to stay as general as possible in terms of expressions and explanations:

*'In Eq. 13, P_f (t) is the probability of failure at time t. Commonly, this problem is referred to with a function named limit state function (g (x, t)), and the safe region is where this function is positive. Thus, the probability of failure would be the'*

Figure 2

The uncertainty of DEL is modelled by log(DELlifetime) ? add description of the uncertainty modelled by DELlifetime . How is this uncertainty quantified and does it include model uncertainty in estimating the stress ranges (obtained from a validation process)? This stochastic modelling assumes that strain gauge measurements are available for the fatigue detail considered?

We believe the reference to is figure '2' incorrect and thus, our answer is according to the general approach with assumption of reference to figure 6. The answer to all questions is 'yes'. An explanation is now added to the descriptions of figure 6 (lines 401– 404) as below:

*'The uncertainty of log(DELlifetime) in the site is modelled by a frequentist approach (Maximum likelihood ) based on observations in the measurements and includes all sources of uncertainty. However, in the case of the other two approaches, the uncertainty of this parameter is assessed based on bootstrapping and thus, it only includes epistemic uncertainty. The data in Fig. 6 are normalized by the converged mean of DEL_lifetime obtained above using site measurements.'*

Eq (14)

Where does the time t enter in the limit state equation?

1:

Below statement is added before Eq. 14 for clarity:

'The time is omitted from Eq. 14 for simplicity with the assumption that all variables are referring to a certain time.'

Line 288

Explain how R=10 is used and why R=10 to account for mean stress level?

The SN curve is derived for that kind of loading; thus, the mean stress effect is already included. This explanation is added now to the same line for clarity:

*'We consider R = 10 for fatigue properties (SN curve) of the composite Mikkelsen (2020). Although the variability of data is included as the CoV of such curve, a calibration is added at the end to set the mean value of material strength to a certain level at which target level of reliability is obtained at year 20.'*

In addition, on the load side, we believe that in the flapwise direction, the ratio of mean to ultimate strength of the material is low compared to the relatively higher cycle range values making the effects of mean stress correction negligible in computation of DEL mean and standard deviation (as shown in [1]).

Table 3

How is the mean value calibrated?

As mentioned in the last paragraphs of introduction: '... the material properties are calibrated such that the target reliability level of 3.3 is reached after 25 years based on design class. ' A brief explanation is also added to the table 3 and the introduction as well as beginning of section 3.3 explaning that a factor is multiplied to the mean level of K.

Mean and standard deviation of log (DELlifetime) are missing in the table?

CoVs of all random variables are included now. However, the mean cannot be presented due to confidentiality.

Line 295

'Based on survival in the year before': not correct – reformulate

 Rephrased to 'conditional on survival in the year before'

Figure 2

Add explanation of all symbols in the figure
* * *
[1] Veers, P. S., "Fatigue Loading of Wind Turbines," Wind energy systems: Optimising design and construction for safe and reliable operation, Woodhead Publishing Ltd., Cambridge, UK, 2011.

1:

Added now.

Explanation of the symbols are added now to the figure

Figure 3

Could a Weibull distribution (as used in IEC 61400-1:20+29) fitted to the upper tail be as representative as the distributions considered?

Weibull is used in IEC 61400-1 for 'loads'. We do use the extreme value theory to model the tail of the 'DEL' data (Eq. 10 to 12). However, in general, we need to fit a full distribution as well since after all it is the weighted mean of DEL_10min that is of interest and not the tail. In the current case the Weibull distribution was not the best fit to DEL data.

Line 336

'Extrapolate the distribution to a 30-year return load': figure 3 shows random point in time observations of the turbulence level. Is this distribution used to estimate the load with a return period of 30 years? Or is the load with a return period of 30 years estimated using e.g. a peak-over-threshold technique considering the extreme, statistical independent loads observed during the measurement period (as in DLC 1.1 load extrapolation)? More explanation is needed.

Thank you for your comment. The latter is correct. More detailed explanations are now added in the methodology (section 2.4.3).

And how to use the load with a return period of 30 years for fatigue assessment?

More detailed explanations are now added in the methodology (section 2.4.3).

Line 357

The target annual reliability index in IEC 61400-1 Annex K is 3.3 (and not 3.7 as indicated in some DNV standards – assuming a ductile failure mode)

This is correct- We also keep the same level as the annex to get a fair comparison when using a reliability-based approach. However, the offshore version of the turbine being used here was neglected before and fortunately, the 25 years of lifetime is giving almost 3.3 in the current curves. Corrections are made now to aim for annual reliability index of 3.3 after 25 years.

Figure 7

As mentioned above the IEC and Frandsen models are intended for deterministic design with safety factors, not for reliability analyses. Recommendation: use the same approach for reliability analysis as in papers and reports related to fatigue of welded steel details in wind turbines.

Thank you again for the comment about the deterministic design versus reliability analyses.

Kindly see the below explanation:

1:

*'It must be noted that the Frandsen and IEC turbulence models together with partial safety factors are intended for semi-deterministic design and not for probabilistic design and reliability analysis. However, since the partial safety factors are calibrated based on achieving certain reliability level (to which we are also setting the values for) at the end of the design lifetime, the results are comparable. Such comparisons are presented in the next section.'*

Regarding the second comment:

Because this study focuses on blade-root moments, which are composite materials, it is not clear to us why we should use the same approach as those for fatigue of welded steel details.

Table 4

How is sensitivity defined?

As mentioned in the table description, they are '*importance rank of the random variables*'. Explanation and reference are added in the methodology now for more clarity (lines 322-324).

1:

---

## Referee Report (RR1)

**"Added value of site load measurements in probabilistic lifetime extension: a Lillgrund case study" (Manuscript number: wes-2024-68 - revision)**

Thank you for the revision of the manuscript. I still think that fatigue assessments are an important topic in the context of wind turbines. Furthermore, conducting them probabilistically and incorporating real strain gauge data is innovative. Nonetheless, the manuscript is still not sufficiently structured, explanations are still missing and even some mistakes have not been removed when revising it. As some comments have not been addressed sufficiently, I do not think that another major revision will solve all these problems. Hence, I cannot recommend it for a publication in the WES journal.

Comments:

1)  The structure of the paper must be improved to make clear what the main innovation/topic is. Currently, it seems to be a mixture of "probabilistic fatigue assessment", "validation using real data" and "turbulence modelling".
    - The structure has been improved. Nonetheless, sometimes it remains unclear. In the introduction, three scenarios are mentioned. However, on the one hand, only two of them are used later on (scenario I and III). Scenario II is only used for validation purposes. On the other hand, three groups of simulations are conducted, which could be linked to the scenarios. However, this is not done. Moreover, frequently measurements are mentioned, but sometimes it remains unclear whether it refers to load measurements or site-specific turbulence measurements.

2)  Abstract: It remains unclear what the topic paper is (see comment 1)
    - See comment 1

3)  In my opinion, the title of the paper does not represent in main topic of the work. Perhaps, turbulence modelling can be included in the title.
    - Title has been changed but it still does not represent the topic of the work. What is the added value of load measurements? This is not really discussed.

4)  Introduction: The connection between the assessment using the Frandsen model (simulation-based, l. 24-44) and the limited data (measurement-based; l. 46-50) is unclear.
    - Connection is still not clear. Does the limited data only refer to load measurements or also to turbulence measurements or even to scenario I, where limited simulation data are available?

5)  The state of the art (L. 52-67) is not sufficient and does not clearly differentiate between simulation-based and measurement-based approaches.
    - Has not been addressed

6)  L. 121: Where exactly is the met mast situated? Please, show it in Figure 1.
    - Done

7)  L. 121: Are shadow effects of the met mast considered, e.g., reduced wind speeds if the anemometer lies behind the met mast.
    - Done

8)  L. 121: At which height(s) is the wind speed measured?
    - Done

9)  L. 124: Your data is biased, as you only cover periods in the winter/spring. This should at least be discussed. Is this bias relevant for your work?
    - Done

10) L. 131: How much data has been removed?
    - Done

11) Table 1: It is not clear for which time period the wind direction bin probabilities are given. Are these the probabilities for the same five years? And are they used somewhere. If yes, please highlight it. If not, you might just remove them.

- It has not been answered for which time period the probabilities (according to Vitulli et al.) apply. Furthermore, it remains unclear which probabilities are used for Eq. (8).

12) Section 2.3.2: Your measurements come from an offshore turbine. The simulations seem to be done for an onshore turbine or all details regarding the offshore part are missing. Just simulating an onshore turbine and comparing it to offshore measurements does not seem to be sensible, even if you focus on blade loads.

- Even though there is some validation in the appendix (which should be in the paper itself as it is quite important), I still do not believe that you can directly compare the results of a simulated generic onshore wind turbine with an offshore wind turbine. I think that this is one reason why the reliability indices in Fig. 7 are so different.

13) L. 159: The site-specific turbulence distribution is not given, but only the reference turbulence intensity.

- Why are the distributions not given?

14) L. 162: How has the exponent of 0.1 been determined using in-situ measurement data?

- Fine, but the values in Table 2 are wrong. 0.1 for group 1 and not for group 3.

15) Table 2: Why are the cut-in, the rated and the cut-out wind speed different compared to the real turbine (Section 2.1)?

- Done

16) L. 174: For groups 1 and 2 you use Rayleigh distributions (covering wind data of full years) whereas the biased measurement data (see comment 9) is used for the strain gauge-based approach. Hence, a direct comparison, as in Figure 7 is not possible.

- I understand that the measurement data is not biased. Nonetheless, in the end, you compare annual reliabilities for group 1 and 2 (where the probabilities are determined using the Rayliegh distribution) and reliabilities for scenario III, where you use the actual data. Hence, the bin probabilities are different. Therefore, you cannot compare these cases directly. This might be another reason for the large differences in reliability indices in Fig. 7.

17) Eq. (3) and (4) are not sufficiently explained, e.g., $di(\theta)$

- Done

18) Section 2.4.2: Formatting and explanations are not sufficient, e.g., $I_y$ and not $Iy$, $N_s$ is not explained etc.

- Formatting is still not completely correct

19) Eq. (8) and (9): At the left side of the equation, the expectation E has to be removed, as $DEL_{lifetime}^m = E(DEL_{10min}^m)$ and not $E\left(DEL_{lifetime}^m\right) = E(DEL_{10min}^m)$

- Done

20) Eq. (9): Index i is missing.

- My mistake.

21) L. 240 and l. 247-264: For me, it is not clear, why we need all this. If I understand it correctly, you fit a distribution to the 10min values (step 1). Then, you sample from this distribution to determine the lifetime value (step 3 and 4). Why do we need the DELs with long return periods. A single DEL with a high return period does not influence the overall lifetime DEL. Hence, they are not relevant and actually not used for the reliability assessment in Section 2.4.4.

- I understand that the high DEL values have an influence on the lifetime DEL. Still, the relevance and even the execution of step 2 is not clear. You could immediately sample (step 3) from the distribution (step 1). However, you somehow form a database (step 2) using the extreme values. Does this mean that you determine a "new" distribution based on Eq. (10) to (12)? In this case, the question is how valid this distribution is, since Eq. (10) is only valid for the tail of the distribution.

22) L. 245: You neither show the fitted distribution for the lifetime DEL nor you state what type of distribution it is.
   - Done
23) Eq. (10) where does this equation come from? It does not exactly match with Eq. (12), which is frequently used in literature.
   - Done
24) Eq. (11): This equation is wrong, as it gives negative probabilities, since the CDF is always between 0 and 1.
   - Done
25) Eq. (14) to (17): Please, revise these equations, as they are not always correct, formatting has to be improved and explanations are missing, e.g., $\Delta t$ and $P_f$ are not explained, it has to be $I_y$ and not $I$, the left side of Eq. (16) has to be $\Delta P_f(X, t + \Delta t)$, $m$ not $R$ etc.
   - Partly done. You still use $K$ and $k$; it is not stated that $R$ is the stress ratio; $c$ is defined as the diameter, but in line 221 it is the radius.
26) L. 289: Why do you apply FORM and not MCS? Your limit state function can be evaluated computationally efficiently, so that MCS should not be a problem and MCS is more accurate.
   - I am not convinced, but it is fine for me.
27) L. 308: How do you define "enough data"?
   - Done
28) L. 313: You state that the Frandsen model and the ICE design underestimate the turbulence for low wind speeds and overestimate it for high wind speeds. I cannot see this in Figure 14, e.g., the Frandsen model is above the 75% quantile for 4m/s and below the same quantile for 20 m/s.
   - The discussions about the Fig. 2 are still not correct, e.g., "the Frandsen model estimations are higher than design in high mean wind speeds" → see bin 1, 20m/s: circle (design) lies above square (Frandsen)
29) L. 330: Why do you investigate this type of multi-modal distributions and not others?
   - Done
30) L. 334-344 and Figure 4 and 5: Why do we need this? For Section 3.3, it is not needed.
   - I understand that Figure 4 is useful (although I still do not understand step 2 (see comment 21)). Figure 5 just shows the convergence for higher $N$. This is not really needed. Perhaps, it is useful in the appendix. However, in the paper, you could just use a high $N$, as you finally did.
31) L. 336: You state that "the probability of the largest data observed" corresponds to five years. However, this is not correct, since you do not have data of five full years.
   - Even if your data does include measurements in all months for five years, this does not lead to a return period of five years. Only if you have measured the highest DEL by chance, it is actually the five-year extreme. In all other cases, it is below. How much below, you do not know. Statistically, it is probably around a two-year return period, as the amount of data sums up to two full years (line 129)
32) Table 4: How did you determine the sensitivities?
   - My fault, you did mention it
33) Table D1: How are the parameters of the different distributions defined?
   - I asked about a definition, not about the determination of them. This is important, as, for example, Par1 and Par2 in a uniform distribution could be the lower and the upper limit, but also the mean value and the standard deviation, etc.

Typos etc.:

34) L. 69: "assess" not "assesses"
   - Done

35) L. 86 and others: "Section" and not "Sect." or "section". Same applies to "Eq.", "Table" etc. Or at least be consistent.
    - Not done everywhere, e.g., line 139 and 505, "table" and not "Table".
36) L. 133: "in Table 1" not "in 1"
    - Done
37) L. 138: I think it is "D1" and not "D2". Overall, reference to figures in the appendix are not always correct.
    - Done
38) L. 174: "Rayleigh" not "Reighley"
    - Done
39) L. 241: $365 \times 24 \dots$ not $365 * 24 \dots$
    - Done
40) L. 346: "in Fig. 6" not "in 6"
    - Done
41) Figure 6: $I_{ref}$ not Iref
42) Table 4 (and appendix): Do not use the notation 7.62e-3, but $7.62 \times 10^{-3}$
    - Not done in the appendix
43) L. 392: "fatigue" not "Fatigue"
    - Done
44) L. 419: "h and more"?
    - Done
45) L. 446: "In the following sections, we compare the turbulence levels in three scenarios of the study"?
    - Done
46) Caption of Table D1 has to be corrected.
    - Done
47) Caption of Figure D2 has to be corrected.
    - Done

Some other formatting errors and typos are now present due to the revision:

- Missing brackets when citing, e.g., line 19 or 23
- L. 24: "re assess"
- "Aero-elastic" in line 30 but "aeroelastic" in line 33
- L. 209: "wöhler" instead of "Wöhler"
- L. 229 and 230: "Cyclic loading"?
- …

---

## Referee Report (RR3)

**Peer Review Report**

**Title: Added value of site load measurements in probabilistic lifetime extension: a Lilligrund case study**

**Summary**

The revised manuscript addresses several of my previous comments and shows meaningful improvement in clarity and structure. The addition of Figure 1 and clearer scenario definitions significantly enhance readability. The methodology section is more transparent, and references to standards (DNV-ST-0262) and previous work have been included. Language and grammar have improved noticeably.

**Strengths of Revision**

- Scenarios are now clearly introduced and summarized in a flowchart.

- Aeroelastic model description expanded; digital twin concept clarified.

- Added relevant standards and improved overall readability.

**Remaining Issues**

The paper has improved substantially in clarity and completeness of discussion, but technical depth remains limited in some areas. I recommend minor revision focusing on:

- **Assessment of the Frandsen based results**: The lifetime extension assessment using the Frandsen model is strongly influenced by the performance of the generic aeroelastic simulation model. Therefore, the statement that Frandsen yields conservative results may mean one of the following:

    - It appears conservative **despite the aeroelastic model underestimating loads in freestream conditions**.

    - It appears conservative **partly because the aeroelastic model overestimates loads in freestream conditions**.

    - **Only if the aeroelastic model perfectly matches measurements in freestream conditions** can conservativeness be attributed solely to the Frandsen turbulence estimation.

    Clarifying this would improve the conclusions made in the article.

- **Validation representation:** box plots or summary statistics (at least for plots A1 and B1) would improve credibility and understanding of the aeroelastic model performance (instead of or in addition to these point clouds)

- **Figure 3:** Turbulence measure clarification needed: Is Turbulence given as standard deviation (m/s) of the wind speed for 10-min bins?

---

## Author Response (AR2)

**Response letter to referee #1:**

We would like to thank referee #1 for the comprehensive review and relevant comments. In the first review, we revised the paper based on given comments leading to significant changes in the structure and presentation of the work. We believe that the major comments are addressed previously, and now we have continued to refine based on the second comment letter. Below, are the initial comments that the reviewer believes is still relevant (in black text), their remaining concern about the comment (black text with bullet point) and our response to them (in blue text). The revised version of the paper is presented with traceable changes.

Note: Below, we have not included the comments which the reviewer approves the application of them in the previous round of review.

**"Added value of site load measurements in probabilistic lifetime extension: a Lillgrund case study"** (Manuscript number: wes-2024-68 - revision) Thank you for the revision of the manuscript. I still think that fatigue assessments are an important topic in the context of wind turbines. Furthermore, conducting them probabilistically and incorporating real strain gauge data is innovative. Nonetheless, the manuscript is still not sufficiently structured, explanations are still missing and even some mistakes have not been removed when revising it. As some comments have not been addressed sufficiently, I do not think that another major revision will solve all these problems. Hence, I cannot recommend it for a publication in the WES journal.

**Comments:**

The structure of the paper must be improved to make clear what the main innovation/topic is. Currently, it seems to be a mixture of "probabilistic fatigue assessment", "validation using real data" and "turbulence modelling".

• The structure has been improved. Nonetheless, sometimes it remains unclear. In the introduction, three scenarios are mentioned. However, on the one hand, only two of them are used later on (scenario I and III). Scenario II is only used for validation purposes. On the other hand, three groups of simulations are conducted, which could be linked to the scenarios. However, this is not done. Moreover, frequently measurements are mentioned, but sometimes it remains unclear whether it refers to load measurements or site-specific turbulence measurements.

Thank you for the feedback and the rest of the points. The points are addressed as below:

- The clarification that the study compares two scenarios, and the third one is only used for validation is now added in the introduction.
- There is no one-to-one relation between the simulations and the scenarios and only scenario 2 can be related to group 2 of the simulations. This connection is now elaborated in section 2.3.2.

- Very valid comment. Thank you. All 'measurement' words are not clarified through the text (mentioning which measurement we are referring to).

2) Abstract: It remains unclear what the topic paper is (see comment 1)

• See comment 1

Abstract is now modified with use of clearer (and to the point) wording.

3) In my opinion, the title of the paper does not represent in main topic of the work. Perhaps, turbulence modelling can be included in the title.

• Title has been changed but it still does not represent the topic of the work. What is the added value of load measurements? This is not really discussed.

The difference in the overall reliability level (corresponding to extra additional years of lifetime), are the added value of having measurements. This is more clarified now in the abstract to make the title more understandable. As mentioned before, the turbulence modeling uncertainty and extrapolation methodology are two side results of the main work but not the main objective.

4) Introduction: The connection between the assessment using the Frandsen model (simulation-based, l. 24- 44) and the limited data (measurement-based; l. 46-50) is unclear.

• Connection is still not clear. Does the limited data only refer to load measurements or also to turbulence measurements or even to scenario I, where limited simulation data are available?

With application of comment #1 (regarding referring to the type of measurement in the text), we believe this question is now resolved and the reader would understand what we are referring to.

5) The state of the art (L. 52-67) is not sufficient and does not clearly differentiate between simulation-based and measurement-based approaches.

• Has not been addressed

We agree that there is no relevant literature review for this point of novelty as it is a is a new topic and is not continuing of any previous research. However, for sake of the reader's technical background for the topic we have emphasized on description of scenarios later in methodology instead of referring to unrelated literature. The two challenges which are addressed alongside the main topic (limitations of Frandsen model and extrapolation of loads) have related literature review which is included in the introduction.

11) Table 1: It is not clear for which time period the wind direction bin probabilities are given. Are these the probabilities for the same five years? And are they used somewhere. If yes, please highlight it. If not, you might just remove them.

• It has not been answered for which time period the probabilities (according to Vitulli et al.) apply. Furthermore, it remains unclear which probabilities are used for Eq. (8).

The probabilities are taken from a reference (as also mentioned in the title of table 1). Equation 8 is a general equation and is true for any duration. However, for sake of clarity, we edited the title of the last column in table 1 with emphasizing on the time duration.

12) Section 2.3.2: Your measurements come from an offshore turbine. The simulations seem to be done for an onshore turbine or all details regarding the offshore part are missing. Just simulating an onshore turbine and comparing it to offshore measurements does not seem to be sensible, even if you focus on blade loads.

• Even though there is some validation in the appendix (which should be in the paper itself as it is quite important), I still do not believe that you can directly compare the results of a simulated generic onshore wind turbine with an offshore wind turbine. I think that this is one reason why the reliability indices in Fig. 7 are so different.

The study does not focus on the support structure and only shows the differences regarding fatigue of blades' root (part of rotor nacelle assembly (RNA)). RNA is normally not much affected by the wave loading. However, this limitation of the work is pointed out in the discussion section to suggest future work.

13) L. 159: The site-specific turbulence distribution is not given, but only the reference turbulence intensity.

• Why are the distributions not given?

As distribution of the turbulence has not been used in this research, we do not include this extra information. However, the scatter of measurements is shown in appendix for reader's information.

14) L. 162: How has the exponent of 0.1 been determined using in-situ measurement data?

• Fine, but the values in Table 2 are wrong. 0.1 for group 1 and not for group 3.

Table 2 is correct and shows 0.1 for group 1 and not for group 3. We believe there is a misunderstanding. Please elaborate.

16) L. 174: For groups 1 and 2 you use Rayleigh distributions (covering wind data of full years) whereas the biased measurement data (see comment 9) is used for the strain gauge-based approach. Hence, a direct comparison, as in Figure 7 is not possible.

• I understand that the measurement data is not biased. Nonetheless, in the end, you compare annual reliabilities for group 1 and 2 (where the probabilities are determined using the Rayliegh distribution) and

reliabilities for scenario III, where you use the actual data. Hence, the bin probabilities are different. Therefore, you cannot compare these cases directly. This might be another reason for the large differences in reliability indices in Fig. 7.

The intention is comparing these scenarios exactly as they are performed in lifetime extension assessments nowadays. The difference may be coming from different assumptions; however, the difference is exactly what it is when someone switches from one method to the other. This is the main intension of the current paper: to show how the result can be different.

18) Section 2.4.2: Formatting and explanations are not sufficient, e.g., $Iy$ and not $Iy$, $Ns$ is not explained etc.

• Formatting is still not completely correct

Formatting is corrected.

21) L. 240 and l. 247-264: For me, it is not clear, why we need all this. If I understand it correctly, you fit a distribution to the 10min values (step 1). Then, you sample from this distribution to determine the lifetime value (step 3 and 4). Why do we need the DELs with long return periods? A single DEL with a high return period does not influence the overall lifetime DEL. Hence, they are not relevant and actually not used for the reliability assessment in Section 2.4.4.

• I understand that the high DEL values have an influence on the lifetime DEL. Still, the relevance and even the execution of step 2 is not clear. You could immediately sample (step 3) from the distribution (step 1). However, you somehow form a database (step 2) using the extreme values. Does this mean that you determine a "new" distribution based on Eq. (10) to (12)? In this case, the question is how valid this distribution is, since Eq. (10) is only valid for the tail of the distribution.

The intention is to include the extremes (occurrences with higher magnitude and lower probabilities) in the samples. The effect can be important in high fatigue exponents as the references provided show (kindly see introduction for related literature and the motivation behind).

25) Eq. (14) to (17): Please, revise these equations, as they are not always correct, formatting has to be improved and explanations are missing, e.g., $\Delta t$ and Pf are not explained, it has to be Iy and not I, the left side of Eq. (16) has to be $\Delta Pf(X, t + \Delta t)$, $m$ not $R$ etc.

• Partly done. You still use $K$ and $k$; it is not stated that $R$ is the stress ratio; $c$ is defined as the diameter, but in line 221 it is the radius.

All mentions are using capital version 'K' to be consistent. R clarified in L. 319 of the revised document. Diameter changed to 'radius' in L. 304 of the revised paper and now we are consistently referring to it as radius through the document. Thank you for your comment and attention.

28) L. 313: You state that the Frandsen model and the ICE design underestimate the turbulence for low wind speeds and overestimate it for high wind speeds. I cannot see this in Figure 14, e.g., the Frandsen model is above the 75% quantile for 4m/s and below the same quantile for 20 m/s.

• The discussions about the Fig. 2 are still not correct, e.g., "the Frandsen model estimations are higher than design in high mean wind speeds" → see bin 1, 20m/s: circle (design) lies above square (Frandsen)

Thank you for your attention. The word 'higher' is now changed to 'lower' as it was a mistake in writing. Additionally, the sentence before is edited for sake of correctness.

30) L. 334-344 and Figure 4 and 5: Why do we need this? For Section 3.3, it is not needed.

• I understand that Figure 4 is useful (although I still do not understand step 2 (see comment 21)). Figure 5 just shows the convergence for higher $N$. This is not really needed. Perhaps, it is useful in the appendix. However, in the paper, you could just use a high $N$, as you finally did.

We hope that explanations to the comment 21 can help with providing more elaborations. However, we also would like to add that the relevance of the whole section is that (as mentioned in the introduction and abstract):

 The study is providing general solutions for using short/mid-term data for long-term fatigue assessment. Although in the current study, the bias in not very large between the resulting DEL in 30 years compared to 5 years, the procedure shown is generally valid and useful.

We believe figure 5 helps with understanding the procedure by illustrating the outcomes of it and since this procedure belongs to the body, we have included the figure in the body of results.

31) L. 336: You state that "the probability of the largest data observed" corresponds to five years. However, this is not correct, since you do not have data of five full years.

• Even if your data does include measurements in all months for five years, this does not lead to a return period of five years. Only if you have measured the highest DEL by chance, it is actually the five-year extreme. In all other cases, it is below. How much below, you do not know. Statistically, it is probably around a two-year return period, as the amount of data sums up to two full years (line 129)

The return period is defined by the mean time between occurrences of a load. Although the possibility of not having the 5-year return period is accepted, we cannot agree to the fact that the unconscious duration of data will define the load level. As the measurements are collected through all months with a constant frequency, the possibility of hitting a load that occurs once every 5 years is not low (we agree it is not 100%). Overall, the extrapolation process which is shown here (which is the main aim of the process) would be the same. The explanation of the limitation is now added to the discussion section.

33) Table D1: How are the parameters of the different distributions defined?

• I asked about a definition, not about the determination of them. This is important, as, for example, Par1 and Par2 in a uniform distribution could be the lower and the upper limit, but also the mean value and the standard deviation, etc.

Apologize for misunderstanding. Definitions are now added to the title of the Table for clarification.

**Typos etc.:**

All comments are applied and a comprehensive edit is not and edit proofed by NREL.

**Response letter to referee #2:**

**Below suggestions for minor revision of the paper are provided by referee #2 (black text). We would like to very much thank you the referee for the constructive comments and very valuable insights shared in both the comment letters. Our responses are shown in blue. The revised version of the paper is presented with traceable changes.**

Line 257
The expression '30-year return loads' can be misunderstood – it is extrapolation to a certain quantile. Delete '30-year return loads'?
The same applies to text about 'maximum loads' which is confusing in relation to fatigue loads.
The extreme value theory is used to cover the extremes of DEL_10min. However, in the referenced instance, the wording is now corrected to avoid misunderstandings.

2.4.1
Frandsen model: In the Dr thesis by Frandsen probabilistic models for model uncertainty is described. Should be included

Thank you for your suggestion. The reference added now in the introduction:

'The Frandsen model involves simplified assumptions, and uncertainties. The uncertainty of the model in estimating the resulting fatigue load in a few examples is presented by Frandsen (2007).'

Eq (14)
It is essential that time t is included

Time added as to be equal to lifetime with additional explanation in the text. Thank you for pointing this out.

Table 3
Are COVs instead of std.dev. values shown?
Is it correct that there is (almost) no uncertainty related to log(DELlifetime)? Generally uncertainty related to estimating the fatigue load is the most important source of uncertainty. Typically the model uncertainty related to fatigue load is the dominating uncertainty, especially for high m values.
The mean value of logK is calibrated to obtain the annual reliability index of 3.7? the mean value and std.dev. of logK from the reference stated should be used.

Yes, the coefficients of variations are mentioned as shown in the title of columns.

Correct, the data in the field showed very low variability which is one of the reasons behind higher reliability compared to the simulation results. This is discussed in the discussion (the second paragraph in the discussion). If referring to the model uncertainty for load (i.e. uncertainty of the aeroelastic model or WTG model), we agree about the large effect on variability and uncertainty of the loads. However, the

CoV shown in the table is taken from strain gauge measurement data. If referring to fatigue model uncertainty (i.e. Miner's rule), that is represented by delta in the present work.

The mean of the DELs is confidential, and we are afraid that with providing the mean of the resistance, mean of DEL can be back calculated. Thus, we have avoided presenting the values. We hope this is understandable.

Section 2.4.3 and Figure 3
Why use extreme value theory to fit fatigue loads? A good fit to the upper tail can be expected to be important for estimating the fatigue life. Add more explanation and show the fit derived into Figure 3.

This sentence added to the text before the formulations of extreme value theory for more clarity and emphasize: '*As we are aiming at adding these low probability high magnitude occurrences, extreme value theory can be a suitable model to use.*'

The reasoning for including the high loads is given with reference in the same part of the text.

As for modeling the DEL_10min the focus has been capturing the extremes and extrapolation of them, we assume that showing the fit on the CDF is more relevant than on the PDF of data. Thus, the fit is shown in the exceedance plot (figure 4) and not on the pdf (figure 3). We hope this is acceptable.

Line 326-328: is it correct that the partial safety factors for blades/composites considered are calibrated to achieve certain reliability level? Please add a reference.

Yes, that is correct. Thank you for your suggestion. We added below reference text is added before presenting table 3:

'*The mean of the material resistance is found through calibration. The calibration process entails finding a resistance mean value for which the target reliability is achieved at the end of the design lifetime.*'

---

## Author Response (AR3)

**Response letter to referee #2:**

**Manuscript** WES-2024-68

**Title:** Added value of site load measurements in probabilistic lifetime extension: a Lillgrund case study

**We would like to deeply and truly thank the referee #2 for the very helpful and constructive comments. The views this referee kindly shared helped us gain clarity on how any reader would grasp the research idea and resulted in big improvements in our presentation.**

**The major and minor comments from referee #2 (black texts) are answered in the current response letter (blue text). Additionally, comprehensive comments were kindly opened by the reviewer in the .pdf version of the paper. This .pdf file is attached to the current letter with responses to all open comments in the same file. All comments are considered and addressed and/or responded[1]**

**Major Comments**

- Scenario Clarity and Consistency: In the introduction, 3 scenarios are introduced. The scenarios should be clearly and consistently defined throughout the paper. A table or flow chart summarizing the scenarios, data sources, and methods would improve clarity. For example, group three of the simulations is used as scenario 2. Renaming the simulation groups to e.g. A, B, C might also help. The scenarios should be extended to the common and relevant scenario where the original aeroelastic simulation model is not available, and a generic simulation model must be utilized (as in this paper).

  - This comment was very helpful for improving the clarity of the paper. Thank you very much. Figure 1 is now added. We also have now pointed out the generic model availability in both the flow chart (fig. 1) and the text (introduction bullet points: line #30 to #40 in the updated paper).

- Scenario 2 Not Fully Developed: While three data availability scenarios are introduced, only scenarios 1 and 3 are fully analyzed. Scenario 2 — using measured turbulence as input to simulations — is mentioned but not developed beyond model validation. Including a full assessment under Scenario 2 would strengthen the comparison and provide a more complete picture of the trade-offs between turbulence-based and response-based approaches. Then you could directly compare what the simulation model predicts, and you can evaluate the difference between the measurements and simulations and better judge the uncertainty coming from the generic simulation model. I see this as optional, given the big effort it would take, and it can be examined to some extent considering the next point:

  - Thank you very much for your comment. We do agree with your statement regarding the great impact such a comparison can have on this work. However, there were two limitations due to which we did not go for full site-specific simulations using SCADA data as input.
* * *
[1] Except for a few for which the data is unfortunately no longer accessible (e.g. power curve.).

First, unfortunately the accuracy of measurements (for both the wind speed and turbulence) can be questioned as in some directions the turbulence measured is not the same as the turbine C08 (under study) is experiencing. This is because in some directions there are wakes imposed to C08 which are not occurring in the met-mast location. Additionally, in some wind directions, the turbine is blocking the met-mast, and the met-mast might is exposed to wakes which the turbine is not exposed to.

Second, we only have access to the SCADA for turbine C08, and the power generation data of the other turbines is not in hand, there would be an effect from other turbines (that the turbine is impacted from, but the met mast is not) that can introduce errors/uncertainties.

However, a point is now added to the discussions and will be picked up in future work (using other available case studies).

**For more information:**
After receiving this comment, we decided to perform the full simulations (with only 6 seeds) to check the possibility of capturing a realistic fatigue load.

[Figure]

For this purpose, we took limited samples after fitting Gaussian mixture models[2] to wind speed and turbulence (joint distribution) and running simulations using samples covering the probability space. Below are the results in all different direction bins:

2  Zhang, X. and Natarajan, A.: Gaussian mixture model for extreme wind turbulence estimation, Wind Energ. Sci., 7, 2135–2148, https://doi.org/10.5194/wes-7-2135-2022, 2022.

The x axis in figures above represents mean wind speed (in m/s). Even though the samples taken (on the left side) show a good coverage of environmental input data, the resulting fatigue loads are not representative of the DEL space (from the strain-gauge). All these uncertainties show the effectiveness and benefit of having high accuracy wind data measurements and real time updating model. Having such a combination of continuously calibrated models with SCADA and high accuracy real time measurements can add a very high value and show a fourth reliability curve and a more fair comparison between design and site. This is now pointed out in the discussions and will be picked up in future work (using other available case studies).

The mismatches can be due to some of the reasons below:

- The turbine has experienced lower turbulence than measured (due curtailment of the front turbines, difference in the location of the measurements, etc.).

- Yaw misalignments (not considered in the simulations) make the DEL in flapwise direction lower than in simulation.

- The location of measuring turbulence is different from the turbine experiencing a different flow in some directions.

- The generic model used can introduce errors in the load estimations and thus in the 1Hz DELs.

- The simulations may be closer to reality than the measurements are due to strain-gauge loss of calibrations or errors.

- Uncertainty from Generic Simulation Model: The use of a generic aeroelastic model introduces significant uncertainty. The paper should more thoroughly discuss the limitations of this model and how they may affect the results. A sensitivity analysis or a more detailed analysis (e.g., statistical plots) would help quantify this uncertainty.

We have added references and discussions (lines #460-465 and #508-511) based on this comment.

As there is already previous work dedicated to sensitivity assessment of fatigue loads to model parameters[3] , we decided to refer to that work and add some discussions regarding this important factor. As the generic model structural parameters are given by the OEM and only the controller is adapted (tunned DTU 10MW controller), such discussions could be narrowed down to the yaw misalignment effect in the current work. However, the effect of such uncertainties in lifetime extension assessment (sensitivity of the model errors to environmental conditions), could be different and we have suggested continued research in future on this matter (line # 508).
* * *
[3] Robertson, A. N., Shaler, K., Sethuraman, L., and Jonkman, J.: Sensitivity analysis of the effect of wind characteristics and turbine properties on wind turbine loads, Wind Energ. Sci., 4, 479–513, https://doi.org/10.5194/wes-4-479-2019, 2019.

**Minor Comments:**

- Missing Reference to Lifetime Extension Guidelines: The paper does not reference any established lifetime extension verification guidelines such as DNV-ST-0262. Including such references would provide context and help position the study within the current state-of-the-art.

    - Thank you for your comment. The reference to this standard is added now.

- Extrapolation Justification (L71): The authors state that the best method for extrapolating fatigue loads remains elusive yet proceed with a specific approach. Why was this method chosen despite the uncertainty?

    - Thank you for sharing your question. an explanation provided for what has been done in connection with prev. work: The current work presents a procedure capturing multimodality of fatigue load data—as demonstrated to be effective in 'X'—and extrapolates it over the full assessment duration.

- Language and Grammar: The manuscript would benefit from a language review to improve clarity and readability.

    - Proofreading and polishing of the final text is done now. Thank you for including the editorial comments.

- Explanation of Aeroelastic Model: More detail is needed on the aeroelastic model used, including assumptions (it seems that it is an onshore model), controller implementation.

    - Model is presented in more detail now both in the relevant subsection (2.3.2) and the discussion (effects of the specifics on the results are also discussed through lines #460-465).

- Flowchart Suggestion: A visual representation of the analysis process—from data collection to reliability assessment—would greatly aid reader understanding.

    - Included in fig. 1 now.

- DEL Distribution Sensitivity: The quality of the DEL_lifetime estimate depends heavily on the fitted DEL_10min distribution. Please discuss how well the chosen distribution fits the data and whether alternatives were tested.

    - Gaussian mixture model was also tested as another common mixture model (only bimodal distributions were considered due to the shape of pdf).

- Digital Twin: The term digital twin is used sometimes in the paper. Please clarify whether you consider the aeroelastic simulation model as digital twin and if so, what makes it a digital twin.

    - No, we do not have a digital twin in hand. There is now a more detailed description of this word both in the ending lines of abstract and conclusion:
    - '... accurate environmental and load coupled with accurate models updating in real time with load measurements (digital twins)

---

## Author Response (AR4)

**wes-2024-68:** Added value of site load measurements in probabilistic lifetime extension: a Lillgrund case study

Author response to the reviewer comment letter:

We truly appreciate the initial detailed comments from the reviewer. These comments not only led to pronounced improvements in the work and its presentation but also made our view wider on future possible work. Below are the remaining minor comments of the reviewer (in black) with our replies (in blue). All the comments are also addressed in the final revision of the paper.
* * *
The paper has improved substantially in clarity and completeness of discussion, but technical depth remains limited in some areas. I recommend minor revision focusing on:

• **Assessment of the Frandsen based results:** The lifetime extension assessment using the Frandsen model is strongly influenced by the performance of the generic aeroelastic simulation model. Therefore, the statement that Frandsen yields conservative results may mean one of the following:

- It appears conservative despite the aeroelastic model underestimating loads in freestream conditions.
- It appears conservative partly because the aeroelastic model overestimates loads in freestream conditions.
- Only if the aeroelastic model perfectly matches measurements in freestream conditions can conservativeness be attributed solely to the Frandsen turbulence estimation.

Clarifying this would improve the conclusions made in the article.

Thank you very much for your comment. The statement highlighted in yellow is true in this case. This is now clarified in the discussions (lines #463-464) and conclusion (lines #529-530).

• **Validation representation:** box plots or summary statistics (at least for plots A1 and B1) would improve credibility and understanding of the aeroelastic model performance (instead of or in addition to these point clouds)

Table A1 represents the statistics mentioned. Unfortunately, the full data is no longer available for addressing other forms of statistics. However, a clear reference to this table is now made in line #464 in the discussion to address both this comment and the previous one.

• **Figure 3: Turbulence measure clarification needed:** Is Turbulence given as standard deviation (m/s) of the wind speed for 10-min bins?

Yes. This is clarified by including the unit (m/s) both in the figure's caption and in the 'y' axis label.

Additional to the changes asked for by the reviewer, a full final review has been conducted resulting in some refinement and corrections in the text. In addition:

- A point is added to the discussion about the uncertainty of fitting (lines #478 and #513).
- Two descriptive sentences are added in lines #446 as well as #459.

A tracked-change pdf of the article is attached showing all changes.